# Arctic Ocean Sea Level Record from the Complete Radar Altimetry Era: 1991–2018

**Stine Kildegaard Rose** [1],*,†, **Ole Baltazar Andersen** [1], **Marcello Passaro** [2] ,
**Carsten Ankjær Ludwigsen** [1] and **Christian Schwatke** [2]

[1]   Technical University of Denmark—National Space Institute (DTU Space), 2800 Kgs. Lyngby, Denmark
[2]   Deutsches Geodätisches Forschungsinstitut der Technischen Universität München (DGFI-TUM), 80333 Munich, Germany
*   Correspondence: stine@space.dtu.dk; Tel.: +45-45259742
†   Current adress: Department of Geodesy, DTU Space—National Space Institute, Elektrovej Build. 228, 2800 Kgs. Lyngby, Denmark.

**Abstract:**   In recent years, there has been a large focus on the Arctic due to the rapid changes of the region. Arctic sea level determination is challenging due to the seasonal to permanent sea-ice cover, lack of regional coverage of satellites, satellite instruments ability to measure ice, insufficient geophysical models, residual orbit errors, challenging retracking of satellite altimeter data. We present the European Space Agency (ESA) Climate Change Initiative (CCI) Technical University of Denmark (DTU)/Technischen Universität München (TUM) sea level anomaly (SLA) record based on radar satellite altimetry data in the Arctic Ocean from the European Remote Sensing satellite number 1 (ERS-1) (1991) to CryoSat-2 (2018). We use updated geophysical corrections and a combination of altimeter data: Reprocessing of Altimeter Product for ERS (REAPER) (ERS-1), ALES+ retracker (ERS-2, Envisat), combination of Radar Altimetry Database System (RADS) and DTUs in-house retracker LARS (CryoSat-2). Furthermore, this study focuses on the transition between conventional and Synthetic Aperture Radar (SAR) altimeter data to make a smooth time series regarding the measurement method. We find a sea level rise of 1.54 mm/year from September 1991 to September 2018 with a 95% confidence interval from 1.16 to 1.81 mm/year. ERS-1 data is troublesome and when ignoring this satellite the SLA trend becomes 2.22 mm/year with a 95% confidence interval within 1.67–2.54 mm/year. Evaluating the SLA trends in 5 year intervals show a clear steepening of the SLA trend around 2004. The sea level anomaly record is validated against tide gauges and show good results. Additionally, the time series is split and evaluated in space and time.

**Keywords:** radar altimetry; satellite altimetry; arctic ocean; remote sensing of the oceans; sea level rise; polar area

## 1. Introduction

The Arctic region has warmed faster than any other parts of the Earth, where the sea level of the Arctic Ocean is an important climate indicator. The arctic sea-ice is decreasing, and has since 1997 experienced a steepening in the decrease [1]. In the fifth Intergovernmental Panel on Climate Change (IPCC) report a global total sea level rise of $2.8 \pm 0.7$ mm/year in the period of 1993–2010 is found [2]. The sea level rise is due to: (1) Thermal expansion [3]. 93% of the atmospheric energy imbalance, which is caused by greenhouse gases, accumulates in the ocean as ocean heat content. Recent models show an increasing ocean warming trend in the upper 2 km of the oceans [4]. (2) Land water storage from human interactions i.e., ground water depletion and reservoir storage [5]. (3) Glacier and ice

sheet mass losses. Outlet glaciers are losing mass more rapidly [6,7], contributing to the sea level rise, changing the oceans freshwater flux, and influencing the ocean thermohaline circulation [8].

The polar oceans are often not included in the global sea level estimations and can be seen as white spots on the global sea level maps. This is because of the challenging polar sea level determination due to; the seasonal to permanent sea-ice cover, the lack of regional coverage of satellites, satellite instruments ability to measure ice, insufficient geophysical models, residual orbit errors and retracking of satellite altimeter data.

The sea-ice cover is in constant change. The sea-ice extent is the largest in March and the smallest in September. The Norwegian and Barents Sea are only seasonally covered by sea-ice while the central part up to the Canadian Archipelago and the North coast of Greenland are permanently ice covered (see Figure 1 for an Arctic Ocean overview). The older ice is pushed against these parts, and additionally, the Canadian Archipelago and the land-fast ice areas are also the part with the fewest leads and consequently the most inaccurate sea level determination [9].

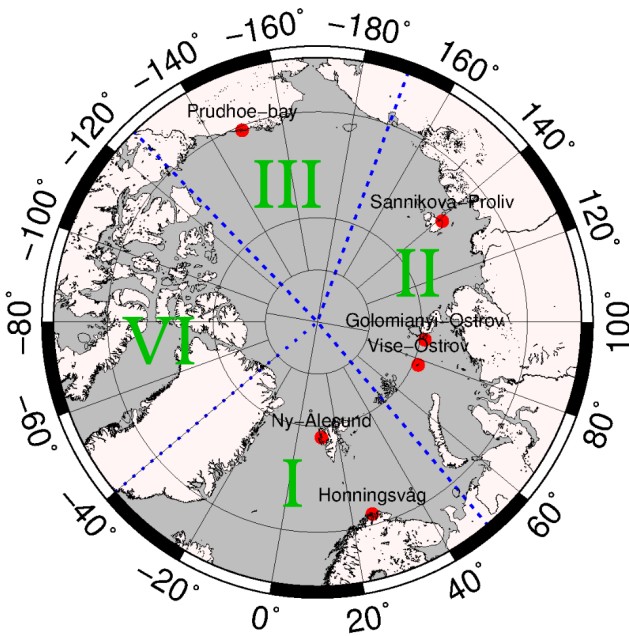

**Figure 1.** Overview map of the Arctic Ocean. The map show the tide gauges (red dots) used to validate the SLA and the different sectors (divided in blue punctured lines) used to investigate the different Arctic regions. The four regions are: I: Fram Strait, Greenland Sea, Norwegian Sea and Barents Sea. II: The Russian Arctic: Kara Sea and Laptev Sea, III: East Siberian Sea and Beaufort Sea, IV: The Canadian Archipelagos and Baffin Bay.

Sea-ice affects the returned satellite radar signal (or waveform) resulting in a poorer coverage and a lower quality of the return signal. Sea level estimates in the sea-ice covered areas are dependent on gaps in between ice floes (leads or polynyas). From now on we are not separating between leads and polynyas but referring to leads as ocean water surrounded by frozen ice. Leads are often very flat ocean surfaces, where there are almost no scatter from the radar wave. This will be registered as a very peaky waveform in the received echo. If leads are located off-nadir their strong backscatter can substantially decrease the quality of the range retracking, this is also known as snagging [10]. In case of Envisat, the nominal circular footprint of 2 km in diameter [11] can increase up to 10 km [12] for strong off-nadir backscatter sources. Despite its much smaller along-track footprint (1.65 km × 0.30 km), CryoSat-2 can also be affected by off-nadir leads, which will result in erroneous range estimates [13]. Refrozen leads are often seen as normal specular lead waveforms but they can be biased up to a couple of centimeters. Another source of errors can be melt ponds on-top of the ice in Spring/Summer and ice freeze-ups in Autumn.

Also the geophysical range corrections are less accurate in the Arctic due to the sea-ice contamination of the radiometer and the lack of observations. The tides and the inverse barometer effect (IBE) are the most important parameters, but also the most uncertain [9,14].

The satellite altimeter era has completed more than 25 years of measurements distributed over several satellites: ERS-1, ERS-2, Envisat, CryoSat-2, ICESat-1/2, SARAL and Sentinel-3A/B. All these satellites have a geographical coverage suitable for Arctic Ocean research. The T/P and Jason satellite series have proven worthy for mid latitude SLA studies, but do not cover the Arctic region. In this study, we use data from four ESA radar altimeter satellites; ERS-1, ERS-2, Envisat and CryoSat-2. In the earlier altimeter satellite missions (ERS-1, ERS-2 and Envisat) orbit errors up to 5 cm still exists [15].

The Arctic Ocean is lacking in-situ measurements consistent in time and space, mainly due to its harsh environment. Various publications (e.g., [16,17]) of the Arctic sea level from tide gauge data exists, where the sea level are measured along the Russian and Norwegian coasts. There exist few tide gauges in the interior of the Arctic Ocean, and they are all short time series. Several bouys have been deployed in the Arctic Ocean ex. Argo (www.argo.ucsd.edu), Ice-Tethered Profiler (www.whoi.edu /website/itp), UNCLOS and GreenArc [18] The Argo buoys have shown great results in validating altimeter data [19], but are not yet densely deployed in the Arctic Ocean.

The first Sea Surface Height (SSH) studies covering large parts of the Arctic Ocean were computed from the ERS satellites to produce sea-ice thicknesses [20] and gravity anomalies [21]. Peacock and Laxon [22] were the first to construct an ocean product, a mean SSH (MSS), from the Arctic Ocean using ERS-1 and ERS-2 altimeter data from a 10 year period. Since then several e.g., [23–26] have followed. Global MSS products are available from: CNES/CLS (not covering the Arctic) [27], DTU [28], SSALTO/DUACS by AVISO.

In this paper, as part of ESA's Sea level CCI (SL_CCI) and the Sea Level Budget Closure (SLBC_CCI), we use 27 years of radar satellite altimeter data for constructing a new improved monthly sea level record for the Arctic Ocean - the CCI DTU/TUM Arctic Ocean data set. We find a sea level rise of 1.54 mm/year of the Arctic Ocean covering 65°N to 81.5°N latitude and $-180°$ to 180° longitude from September 1991 to September 2018, with a 95% confidence interval of 1.16–1.81 mm/year. The coverage from the ERS-1 satellite is sparse during periods of time and the time series may be more error prone in this period, therefore looking at the time series starting from ERS-2 we get a sea level rise of 2.22 mm/year with a 95% confidence interval within 1.67–2.54 mm/year.

The paper starts by describing the data used (Section 2). We use a combination of tailored level-2 (L2) ERS-1 data together with a new retracking of ERS-2 and Envisat data and with a combination of state-of-the-art altimeter data and retracked data from CryoSat-2. In Section 3, the methods are described in making the SLA product from pre-processing (Section 3.1) and geophysical corrections (Section 3.2) to handling the sea-ice (Section 3.3) and the intermission biases (Section 3.4) between the different satellites. The SLA product resampling and gridding are described (Section 3.6), and finally, in Section 3.7 a bootstrap analysis is described to evaluate the SLA uncertainties. In Section 4, the results are described and validated. The section starts by showing the resulting SLA uncertainty (Section 4.1). The results are described as regional trends (Section 4.2), inter-annual variability (Section 4.3) and regional variability (Section 4.4). The sea level anomalies (SLA) are validated against six tide gauge stations shown in Figure 1. This is described in Section 4.5. The results are discussed in Section 5 and summarized in the conclusion (Section 6).

## 2. Data

This section describes the data used in this study.

### 2.1. Altimetry Data

The CCI DTU/TUM Arctic SLA contains data from four ESA radar altimeter satellites ERS-1, ERS-2, Envisat, CryoSat-2. ERS-1, ERS-2 and Envisat are conventional altimetry or low resolution mode (LRM) data sets processed with a single processor, while CryoSat-2 consists of three types: LRM,

Synthetic Aperture Radar (SAR) and SAR Interferometry (SARIn), which are processed with different processors. For satellite specific details see Appendix A. In Section 3.2 the geophysical range correction data are described.

### 2.2. Ice Concentration Data

In Section 3.3, we use sea-ice concentration data in separating sea-ice data from ocean data. The sea-ice concentration data are derived in an operational product (after 2015) [29] and a reprocessed product (before 2015) [30] by the EUMETSAT Ocean and Sea Ice Satellite Application Facility. Both products are given as sea-ice concentrations in 10 km Polar Stereographic grids for every six hours.

### 2.3. Tide Gauge Data

In validation of the CCI DTU/TUM SLA data set (Section 4.5), tide gauge data from the Permanent Service for Mean Sea Level (PSMSL) [31,32] are used. The tide gauge data are given as monthly SLAs. Six tide gauges are chosen spread along the coast of the Arctic Ocean (Figure 1).

## 3. Generation of the Sea Level Product

The Arctic Ocean SLAs are computed by the following steps:

1. Pre-processing
2. Adding/removing geophysical corrections
3. Sea-ice concentration data are used to discriminate between the sea-ice cover and the open ocean
4. Threshold criterias are used to separate the leads/open ocean from the sea-ice
5. Inter-satellite biases are determined and corrected
6. Removing outliers
7. Resampling and gridding the data to compute the final Arctic SLA
8. Uncertainty analysis

### 3.1. Pre-Processing

Pre-processing details for the individual satellites are described in Appendix A.

### 3.2. Geophysical Corrections

The geophysical corrections were updated to get a more uniform product, suitable to compare the SLAs in between satellites. Table 1 summarizes the corrections used.

**Table 1.** Data origin and applied geophysical corrections. O, L, LP tides are the Ocean tide, ocean loading tide, long-periodic non-equilibrium ocean tide, LP otide + setide includes the long-periodic ocean tide and the solid earth tide.

|  | ERS-1 | ERS-2 | Envisat | CryoSat-2 [33] |
|---|---|---|---|---|
| Data origin | REAPER L2 [34] | ALES+ [34,35] | ALES+ [35,36] | LARS/RADS [33,37,38] |
| Wet troposphere | ECMWF [39] | ECMWF [39] | ECMWF [39] | ECMWF [39] |
| Dry troposphere | Radiometer/ECMWF [39] | Radiometer/ECMWF [39] | ECMWF [39] | ECMWF [39] |
| Ionosphere | NICO [40]/GIM [41] | NICO [40]/GIM [41] | Doris [36] | GIM [41]/Bent [42] |
| DAC | ERA-Interim [43] | ERA-Interim [43] | ERA-Interim [43] | DAC-ECMWF [44] |
| O, L, LP tides | FES2014 [45] | FES2014 [45] | FES2014 [45] | FES2014 [45] |
| LP otide + setide | Cartwright [46] | Cartwright [46] | Cartwright [46] | Cartwright [46] |
| Pole tide | Wahr [47] | Wahr [47] | Wahr [47] | Wahr [47] |
| Sea state bias | Altimetrics [34] | ALES+ [48] | ALES+ [48] | None/RADS [38] |
| Mean sea surface | DTU18 [49] | DTU18 [49] | DTU18 [49] | DTU18 [49] |

The preferred method for estimating the wet tropospheric correction over the Arctic Ocean is to use modeled data, due to the radiometer contamination by the sea-ice [50]. For most of the satellites a model correction is available. For ERS-1 REAPER data the microwave radiometer wet tropospheric

correction is applied over the ocean if valid or else a model correction is applied. The authors were not aware of a way to see which of the corrections were applied, and therefore it was not possible to change this correction.

We use the FES2014 [45] ocean tide model with loading effects. This model is optimized in the Arctic Ocean compared to previous versions. The tide model is limited in coastal areas resulting in a final data set not defined close to the coast. FES2014 was produced by Noveltis, Legos and CLS Space Oceanography Division and distributed by Aviso, with support from CNES (http://www.aviso.altimetry.fr/).

The atmospheric correction in the Arctic is very important since amplitudes of the signal can reach 1 m, i.e., greater than the SLA signal. Normally in the Arctic, IBE is favored over the Dynamic Atmosphere Correction (DAC) including high atmospheric fluctuations, because of high latitude issues. For consistency, ERS-1, ERS-2 and Enivsat are reprocessed with the DAC ERA-Interim [43] by linear interpolation in space and time. In consequence, this will give more outliers in the data. The DAC ERA-Interim product are computed in the period of 1991–2015, not covering the total CryoSat-2 period. Therefore, the DAC-ECMWF [44] from CLS is here used from the CryoSat-2 GDR product. Various models were tested, and this was proven to be the one closest to the DAC ERA-Interim model.

The applied sea state bias correction for ERS-1 is taken from the REAPER product. For ERS-2 and Envisat the sea state bias is derived from the ALES+ retracker and applied at 20 Hz [48]. For CryoSat-2 only sea state bias for the LRM mode is applied, which is a hybrid sea state bias from the RADS product. For most cases it is fair to ignore the sea state bias in SAR and SARIn mode, such leads are very flat surfaces, where the sea state bias is very close to zero.

The DTU18 MSS was used as a reference [49]. The new MSS from DTU is improved in the central Arctic region and in coastal zones. It has a bias towards recent years sea level heights including three years of Sentinel-3A and eight years of improved CryoSat-2 data.

*3.3. Lead and Ocean Discrimination*

The Arctic Ocean SLA record is derived by separating leads in the sea-ice cover and open ocean according to the different classification of their surfaces. Various sea-ice types can mistakenly be associated with open ocean waveforms. The ocean is separated from the sea-ice cover by the ice concentration grids (Section 2.2). For more details see Appendix B.

Sea-ice and mixed surfaces are removed by using the waveform Pulse Peakiness (PP) and the width of the leading edge. Furthermore, for CryoSat-2, the stack standard deviation is used to identify the leads. For removing erroneous data in the open ocean (that could be data from the ice edge or near the coast), the PP and the backscatter coefficient are used. There exists many variations of the PP formula e.g., [13,22,51,52]. The values used in this study for each satellite are shown in Table 2. All references to the PP (in Table 2 and in text) are described as in [13], which is given by the waveform maximum power received multiplied with the sum of all range bin powers. In parenthesis, PP values are described as in [22] which is the same formulation as [13] but multiplied with a constant of 31.5. This formulation was first used for the ERS satellites. The choice of these threshold values are based on several studies [11,13,22,53,54] and adjusted and evaluated for this study.

**Table 2.** Thresholds used in lead and ocean discrimination. The table columns are PP, stack standard deviation (St. Std), and width for lead discrimination and PP. Backscatter coefficient ($\sigma_0$) for the ocean discrimination. The PP values are described as in [13] and in parenthesis as in [22]. The PP is calculated differently for CryoSat-2 (see the details in the text). The two numbers corresponds to SAR and SARIn, respectively. The width in ERS-1 is from the REAPER product and is the OCOG width, for ERS-2 and Envisat the width is the ALES+ leading edge rising time, and for CryoSat-2 the width is the width of the Gaussian fit.

|  | Lead | | | Ocean | |
|---|---|---|---|---|---|
|  | PP $>$ | St. Std | Width $<$ | PP $<$ | $\sigma_0 <$ |
| ERS-1 | 0.60 (19) | - | 3 | 0.048 (1.5) | 15 |
| ERS-2 | 0.65 (20.5) | - | 3 | 0.048 (1.5) | 15 |
| Envisat | 0.71 (22.5) | - | 3 | 0.048 (1.5) | 15 |
| CryoSat-2 | 0.35/0.25 (11/7.9) | 4 | 0.9 | - | - |

*3.4. Intermission Bias*

To get a seamless transition between conventional altimetry (from ERS-1/2, Envisat, CryoSat-2 (LRM)) and SAR/SARIn (CryoSat-2) altimetry can be error prone, especially in the Arctic due to the different data coverage. SAR altimeter data have much more data over the sea-ice cover, while conventional altimetry are having troubles. Conventional and SAR/SARIn altimetry data sets are covering different regional areas and are processed with different strategies and having different retracking corrections.

For CryoSat-2, the best approach of merging the different satellite measurement types (LRM, SAR and SARIn) has proven to be a detailed study of individual satellite tracks (not shown). RADS data are in LRM while LARS data are covering SAR and SARIn, so no data are overlapping in time. We found a retracker bias between RADS and LARS of −12.9 cm.

The transition between the four satellite missions, the intermission biases were estimated and minimized. The following steps were completed to handle the intermission biases:

1. Monthly medians were calculated for each mission, over the entire Arctic Ocean, covered by the data sets
2. For overlapping mission pairs (either ERS-1 and ERS-2, ERS-2 and Envisat, or Envisat and CryoSat-2) coinciding months (only full months considered) were detected and extracted
3. For each overlapping pair, the median difference was calculated and the data sets were aligned
4. The biases between the satellites are: ERS-1/ERS-2 ∼0.67 m, ERS-2/Envisat ∼0.53 m and Envisat/CryoSat-2 ∼0.03 m

Figure 2 shows the monthly median of each overlapping satellite pair. The Pearsons correlation coefficient of the three satellite pairs ERS-1/ERS-2, ERS-2/Envisat and Envisat/CryoSat-2 gives 0.52, 0.96, 0.95, respectively.

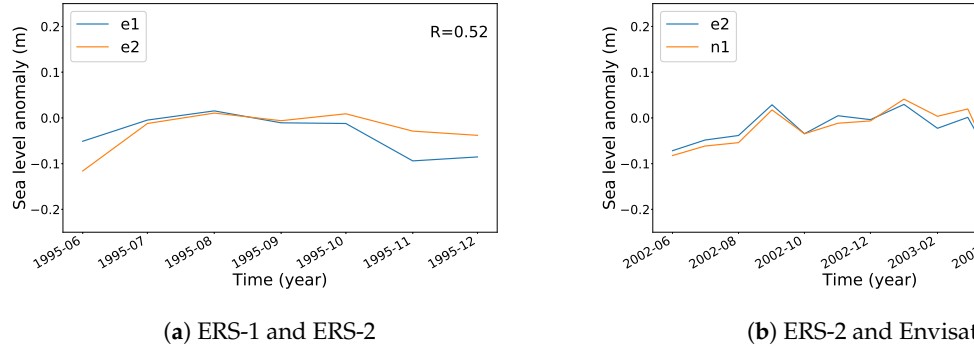

　　　　　　(**a**) ERS-1 and ERS-2　　　　　　　　　　　　　　　　(**b**) ERS-2 and Envisat

**Figure 2.** *Cont.*

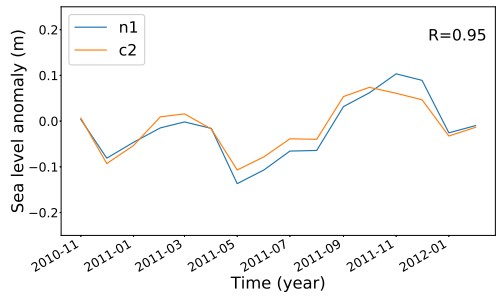

(**c**) Envisat and CryoSat-2

**Figure 2.** Monthly median of the entire Arctic in the overlapping periods for (**a**) ERS-1 (e1) and ERS-2 (e2), (**b**) ERS-2 and Envisat (n1) and (**c**) Envisat and CryoSat-2 (c2). In the top right corner of each figure the correlation coefficient is shown.

### 3.5. Removing Outliers

The outlier removal is carried out in two steps. First, as mentioned in Appendix A, outliers are removed from each track with a MAD outlier detector to get rid of the largest outliers. Second, outliers are detected and removed on a monthly basis with a hard cut-off of $\pm 0.3$ m from the median. This was done similar to Cheng et al. [24]. The hard cut-off resulted in rejection of 18.05% data for ERS-1, 2.45% data for ERS-2, 0.52% for Envisat and 0.06% of data for CryoSat-2. The large removal of ERS-1 data are due to error-prone orbit estimation and bad data sampling, which are causing bad waveforms, resulting in wrong height estimates.

### 3.6. The Arctic Sea Level Anomaly Product

First, monthly data are averaged in cells of $0.2° \times 0.2°$ to overcome the sampling dissimilarity in latitude, which would favor high latitude data especially for Cryosat-2, where the data coverage is much larger than for the conventional altimetry satellites. Second, a least squares collocation with second-order Markov covariance function [55] is used to grid the monthly data. The final grid size is $0.25°$ latitude by $0.5°$ longitude using a 500 km correlation length with a RMS noise of 2 cm. The outputs from the collocation are the SLA data record and a interpolation error estimate both given in monthly grids from September 1991 to September 2018, covering $65°N$–$81.5°N$ and $180°W$–$179.5°E$ in gridline registration. The mean SLAs are shown in Figure 3 for each satellite: ERS-1 (a), ERS-2 (b), Envisat (c) and CryoSat-2 (d). The mean SLA is slightly higher over the sea-ice cover. This is especially the case for ERS-1. For ERS-2 and CryoSat-2 low SLAs controls the Canadian Arctic and the Beaufort Gyre areas, while we see a large mean SLA for Envisat in the Kara Sea. The SLAs in the Fram Strait and Barents Sea areas are slightly negative for the three first missions, while it is slightly positive for CryoSat-2. These figures will be discussed further in Section 5.1.

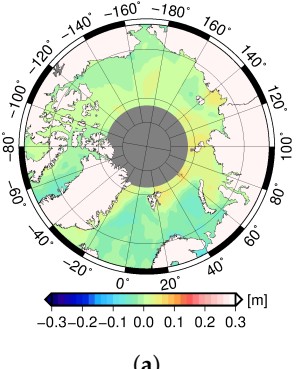

(**a**)

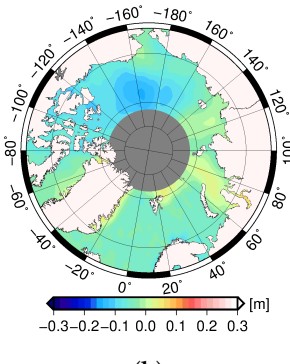

(**b**)

**Figure 3.** *Cont.*

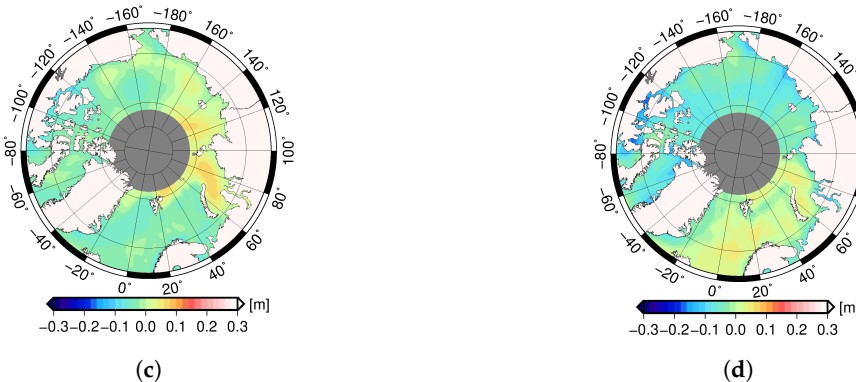

**Figure 3.** The average SLA in meters for each satellite period: (**a**) ERS-1, (**b**) ERS-2, (**c**) Envisat, (**d**) CryoSat-2.

### 3.7. Uncertainty Estimates

In Section 1, multiple error sources that contribute to the total uncertainty of the derived SLA are introduced. These are errors on the altimeter instruments, the orbit determination, the retracking of the radar signal, and from this follows the many uncertainties on the geophysical range corrections (Section 3.2). We can now calculate the true SLA including noise. The exact size of this noise coming from the uncertainties described above are not known, but Ablain et al. [56] looked into this error budget. On top of all these uncertainties there can be errors in the discrimination of ocean and leads (Section 3.3), inter-satellite biases (Section 3.4), in making of the SLA grids (Section 3.6), in making of the total SLA time series and trend maps (Section 4.2), and furthermore, uncertainties can arise from: retracker biases, interpolation, filtering, sampling. The size of all these individual uncertainties are, however, not well known, and additionally it is difficult to propagate the uncertainties analytic in the long processing chain. As an alternative we apply a bootstrap approach [57] to estimate the error of the SLA. Bootstrapping embrace all the variations from the various uncertainties. To obtain valid error estimates using bootstrap, the observations must be independent and the bootstrap data sets must resemble the original data set. Hence, to better approximate independent observations, a block bootstrap is used.

The specific bootstrap procedure to derive the error for each monthly data set is carried out as follows: (1) the data are split in $n$ non-overlapping blocks. (2) 1000 bootstrap realizations are created, by sampling with replacement among the blocks. (3) For each bootstrap data set the SLA is derived in the same way as described in Section 3.6. (4) Finally we have 1000 estimates of the SLA for each grid cell from which we can extract error information such as standard deviation and confidence interval. In Appendix C a more thorough review of the bootstrapping procedure is described.

It is only valid to show results with a standard deviation if the results are normal distributed. The Arctic SLA distributions are not normal distributed for all grid cells in the Arctic Ocean (see Figure A1 in Appendix C for more details). Therefore, the uncertainty is expressed in a 95% confidence level. The results are shown with the median and not the mean value, because of the skewness of the distributions (Figure A1, Appendix C).

## 4. The Arctic Sea Level Anomaly Record

The resulting CCI DTU/TUM Arctic SLA product is analyzed in this section. The SLA product is given by monthly grids from September 1991 to September 2018. These grids are available at DTU: https://ftp.space.dtu.dk/pub/ARCTIC_SEALEVEL/DTU_TUM_V3_2019/ and at ESA SLBC_CCI: (cci.esa.int/data).

Firstly, the total uncertainty of the SLA product is shown. Secondly, we investigate the spatial trend patterns over the entire Arctic region. Thirdly, we show the averaged inter-annual variability. Fourthly, the SLA is validated against tide gauges.

### 4.1. Uncertainty of the Arctic Sea Level Product

The total uncertainty of the Arctic CCI DTU/TUM sea level product from the bootstrapping is summarized (Figure 4). This is given by the monthly median ranges of the 95% confidence interval (i.e., the median of the SLA range between the percentiles 2.5% and 97.5%) from September 1991 to September 2018. We see larger uncertainties in the interior of the Arctic where permanent and seasonal sea-ice appears with a SLA range 50–60 mm compared to the ice-free regions with a SLA range of 10–20 mm.

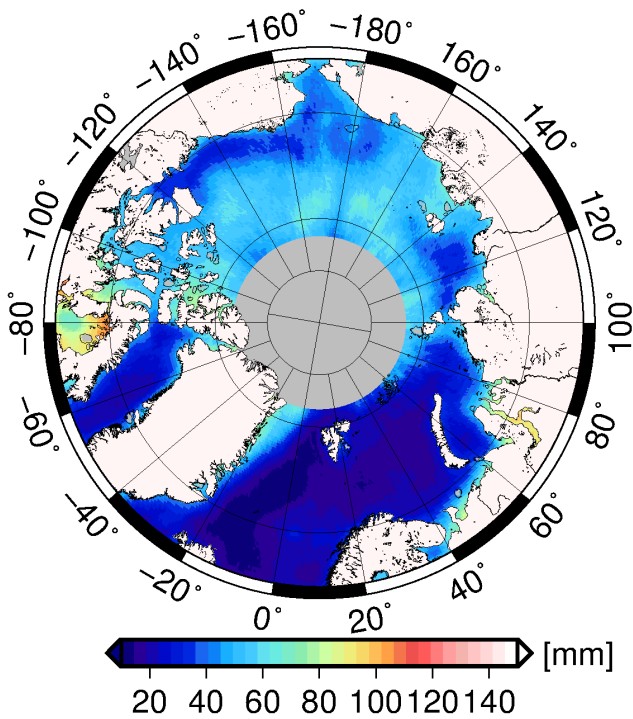

**Figure 4.** The total uncertainty of SLAs from the bootstrapping, given as the median SLA range between the 2.5% percentile to the 97.5% percentile (i.e., the 95% confidence level interval) of monthly data in the SLA product from September 1991 to September 2018.

### 4.2. Regional Trends in the ERS-2 to CryoSat-2 Era

We investigate the spatial trend pattern from 65°N to 81.5°N in the entire Arctic Ocean. In Figure 5a the spatial pattern is shown covering the time period from January 1996 to September 2018. Here, the ERS-1 data are dismissed due to too low data distribution in the Eastern sector. Furthermore, all data are eliminated with an interpolation error (from the collocation) above 10 cm. We find a pattern with a high trend >10 mm/year in the Beaufort Gyre, a slightly negative trend or no trend in the Russian sector (−2 to 1 mm/year), trends between 3–7 mm/year in the Barents Sea and in the Fram Strait, and a strong negative trend in the northern Baffin Bay. The regional trend uncertainties are shown as the 2.5% percentile (Figure 5b) and the 97.5% percentile (Figure 5c) corresponding to the 95% confidence level.

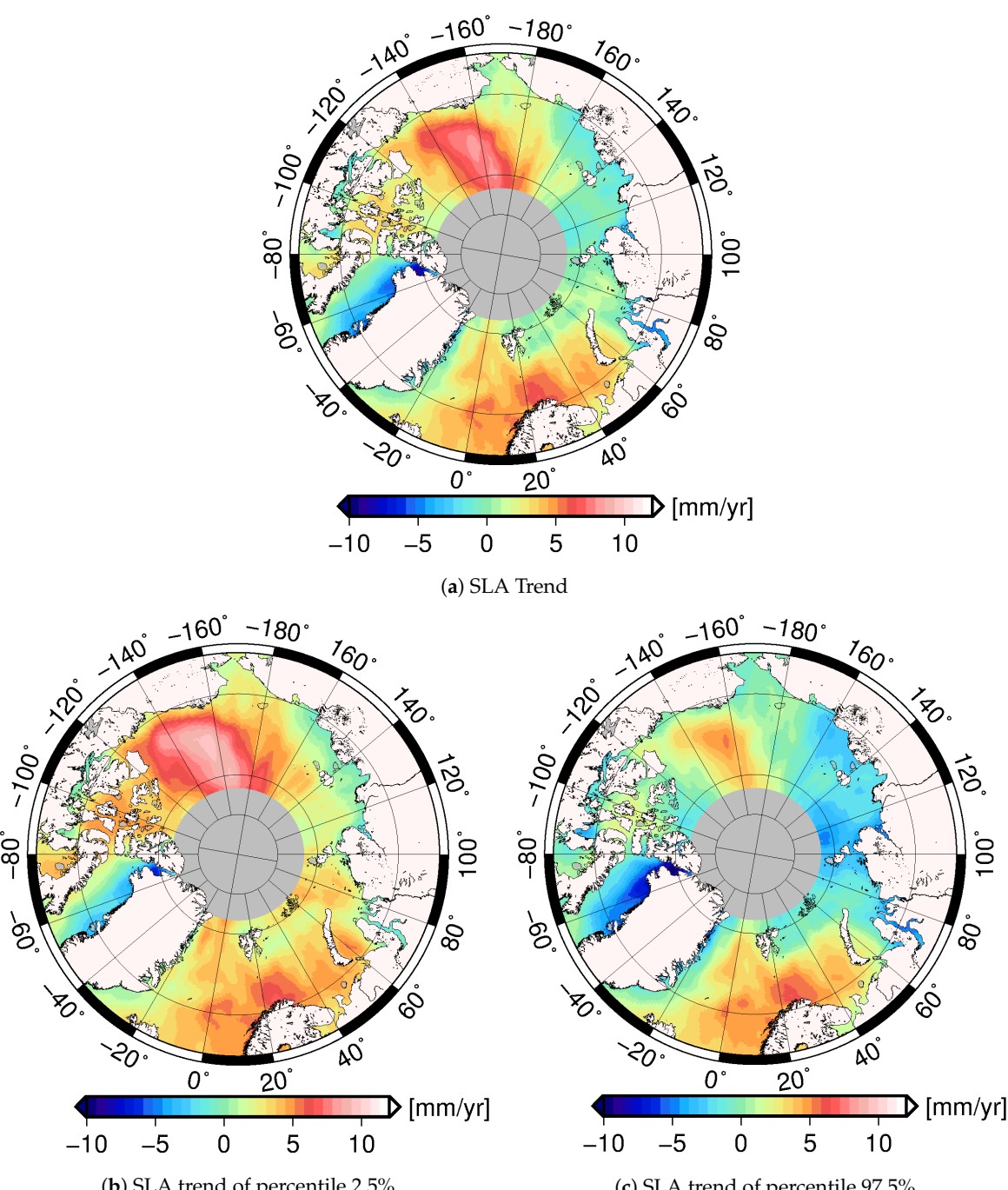

(**a**) SLA Trend

(**b**) SLA trend of percentile 2.5%.

(**c**) SLA trend of percentile 97.5%.

**Figure 5.** (**a**) The CCI DTU/TUM SLA trends from January 1996 to September 2018 given in mm/year. (**b**,**c**) show the SLA trend uncertainty in the same period. There is found a 95% confidence interval of the SLA trend within (**b**,**c**).

### 4.3. Inter-Annual Variability

The SLA data are averaged for each month with a cosine latitude weighting (Figure 6). The seasonal variability are plotted from September 1991 to September 2018 (27 years) in (Figure 6a) and January 1996 to September 2018 (almost 23 year) in (Figure 6b), respectively. We are investigating the time series with and without ERS-1, because the coverage of the ERS-1 satellite is sparse during periods of time (especially in the ice covered regions), and therefore the time series may be more error prone in this period. Both figures show solutions with and without Glacial Isostatic Ajustment (GIA). The applied GIA model is from Caron et al. [58], which is kindly converted to sea level anomalies and

associated standard deviations by Benjamin D. Gutknecht. Generally, we see a seasonal variability of high sea level in late Autumn and a low sea level in the Spring. Both time series have a positive trend with a sea level rise of 1.54 (1.40) mm/year and 2.22 (2.08) mm/year in the respectively periods with and without (in parenthesis) GIA correction). There is a 95% confidence that data lies within 1.16 (1.01)–1.81 (1.67) mm/year and 1.67 (1.52)–2.54 (2.40) mm/year, respectively.

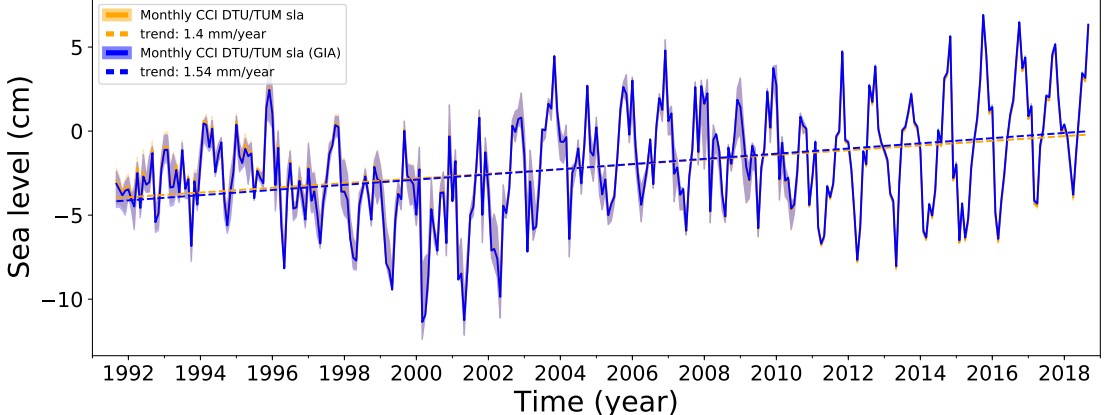

(**a**) Monthly SLA values from September 1991 to September 2018.

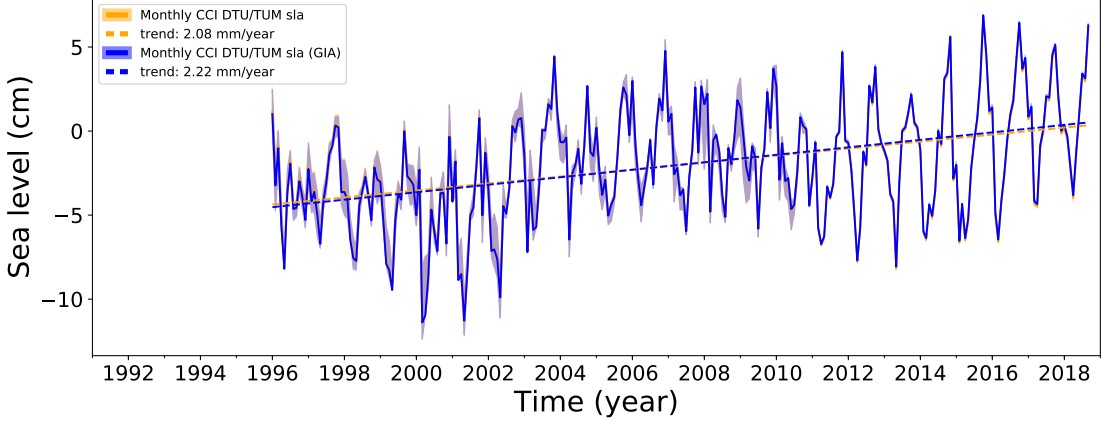

(**b**) Monthly SLA values from January 1996 to September 2018.

**Figure 6.** Monthly SLA values. (**a**) From September 1991 to September 2018 with a linear trend of 1.54 and 1.40 mm/year with a 95% confidence level of data laying within 1.16 to 1.81 and 1.01 to 1.67 mm/year with and without GIA correction, respectively. (**b**) From January 1996 to September 2018 with a linear trend of 2.22 and 2.08 mm/year with a 95% confidence level of data laying within 1.67 to 2.54 mm/year and 1.52 to 2.40 mm/year with and without GIA correction, respectively. The blue and yellow shadows are the 95% confidence level for measurements with GIA and without GIA, respectively.

The uncertainties expressed in Figure 6 as light yellow (no GIA) and light blue (GIA) shadows are derived by continuing each of the 1000 bootstrap realizations through the same procedure as described in Section 3.6. The uncertainties are given as the median SLA range of the 1000 bootstrap realizations in the 95% confidence level.

*4.4. Regional Sea Level Variability*

In Figure 7, the CCI DTU/TUM sea level record from 1996–2018 is divided into four sectors (Figure 1). The regional SLAs with and without GIA and the associated uncertainties are summarized in Table 3.

**Table 3.** SLA trend for each sector and the associated uncertainty. The SLA trend is given with and without GIA correction and the uncertainty is given by a 95% confidence level.

|  | SLA Trend (No GIA) mm/year | 95% Conf. Level (No GIA) mm/year | SLA (GIA) mm/year | 95% Conf. Level (GIA) mm/year |
|---|---|---|---|---|
| Sector I | 3.04 | 2.96–3.23 | 3.19 | 3.10–3.37 |
| Sector II | 0.33 | −0.58–1.28 | 0.04 | −0.86–1.00 |
| Sector III | 4.06 | 2.41–4.71 | 5.77 | 4.12–6.42 |
| Sector IV | 0.49 | −0.72–1.15 | −0.63 | −1.84–0.03 |

Two areas (Sector I (Figure 7a) and III (Figure 7c) have a clear sea level rise in the period. The maximum SLA trend is observed in the Beaufort Gyre (Sector III) up to approximately 10 mm/year. In Sector I (Figure 7a) the highest SLA trend is observed in the southern part towards the Norwegian coast, and smallest along the coast of Greenland and in the upper northeastern part.

Considering the confidence level, Sector II has no or a little positive/negative trend. Sector IV has a positive trend when no GIA is applied, but a negative trend when it is applied. The most negative trend is in the northeastern Baffin Bay of about −10 mm/year. It is unclear if this is due to fresh water flow from the large outlet glaciers or a simple artifact of the LRM to SARIn transition.

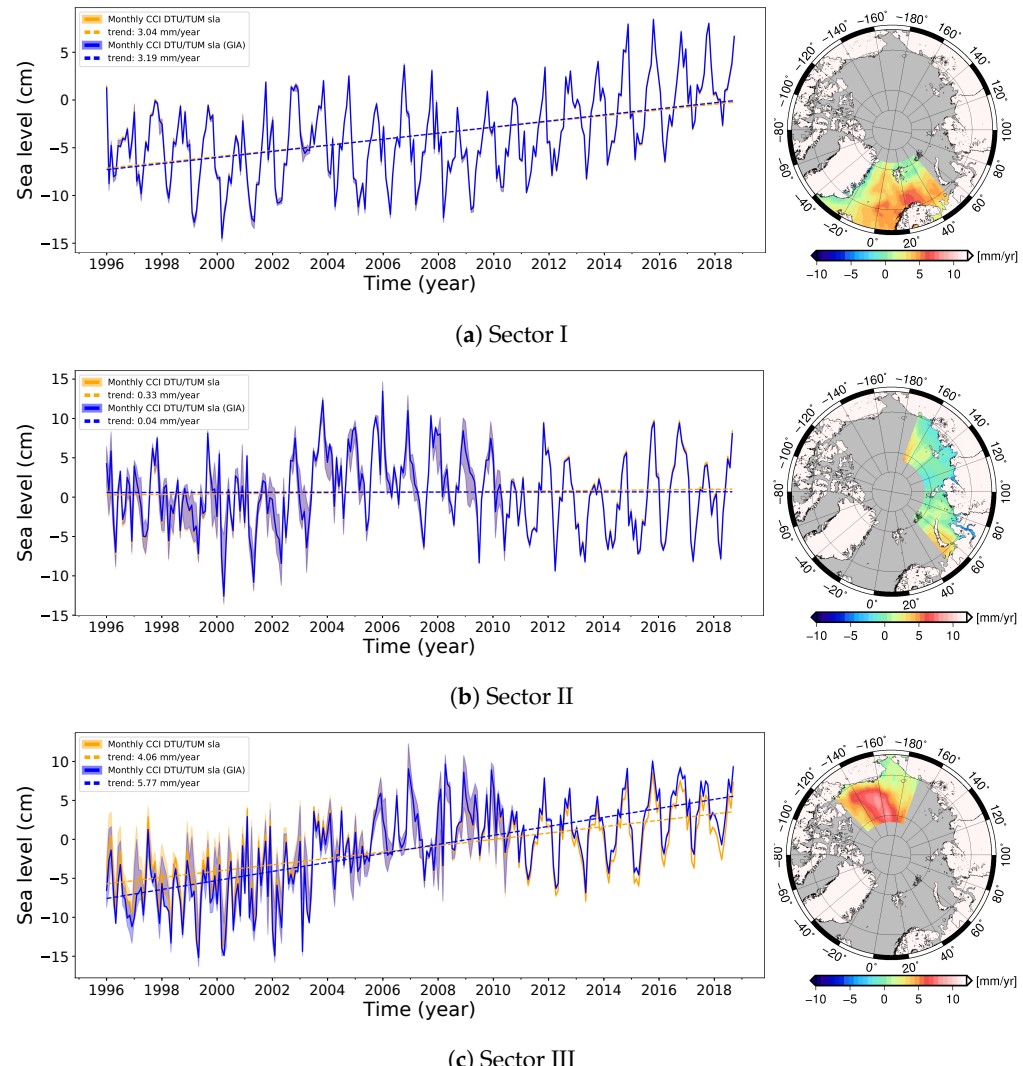

(**a**) Sector I

(**b**) Sector II

(**c**) Sector III

**Figure 7.** *Cont.*

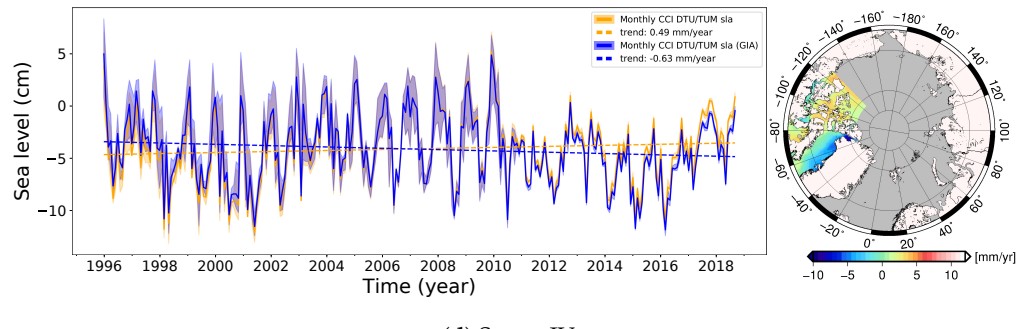

(**d**) Sector IV

**Figure 7.** Regional changes of sea level in the different sectors shown in Figure 1. Blue is the monthly CCI DTU/TUM SLA in meters and the red line is the estimated trend in mm/year for the given sector. The 95% confidence level is given by the blue and yellow shadows with GIA and without GIA, respectively.

### 4.5. Validation

Tide gauges (Section 2.3) in the Arctic are sparsely distributed and gauges with long time series are rare. The six tide gauges used in this study (Figure 1) are chosen due to their geographical distribution in the region, their time span covering most possible of the altimetry era and their continuity in time.

The tide gauges are mounted on land and do not account for GIA effects nor the atmospheric loading. Consequently, in the comparison, the atmospheric loading is not applied to the altimetry data. There is a large GIA signal in the Arctic, with large variations over the region, but GIA models are very uncertain [59,60]. The most direct method for determining the local vertical displacement is by GPS measurements. The GPS vertical displacement includes both the GIA and the elastic signal, whereas the elastic displacement comes from present displacements as mass changes from ex. outlet glaciers or ground water depletion. The elastic displacement is very small if the tide gauge is far from the large mass changes. A vertical GPS displacement is provided when available, or else the closest grid point from the Caron et al. [58] GIA model (given with two times standard deviation) is used (See Table 4).

**Table 4.** Vertical displacement from from GPS and the Caron et al. [58] GIA model.

| Tide Gauge | Vert. Disp. mm/year | GIA mm/year |
|---|---|---|
| Ny Ålesund | 7.98±0.49 [1] | 0.47 ± 0.67 |
| Honningsvåg | 1.9±0.3 [2] | 1.344 ± 0.42 |
| Prudhoe bay | - | −1.51 ± 0.095 |
| Vise Ostrov | - | 1.96 ± 0.38 |
| Golomianyi Ostrov | - | 1.99 ± 0.33 |
| Sannikova Proliv | - | −0.48 ± 0.21 |

[1] Obtained dec. 2018 from www.sonel.org [61]; [2] [62].

The results of the comparison between the CCI DTU/TUM Arctic SLA and the tide gauge data are shown in Figure 8 and in Table 5 and described with more details in Appendix D. We are using an inverse distance weighted average of data in a radius of 350 km from the tide gauge station. Using a radius of 350 km is also done in [24].

Figure 8 compares each satellite relative to the tide gauge. The tide gauge is shown with an orange curve, while the altimetry data are shown with different colors depending on the satellite: ERS-1 (red), ERS-2 (blue), Envisat (green) and CryoSat-2 (Grey). Also trend lines for each satellite are shown for both tide gauges and altimetry data. In Figure A2, Appendix D, the total time series is shown together with the corresponding trend line for every tide gauge station. The trend differences between the altimetry and tide gauge data for Sannikova Proliv (Figure A2e) and Prudhoe Bay (Figure A2f) are large.

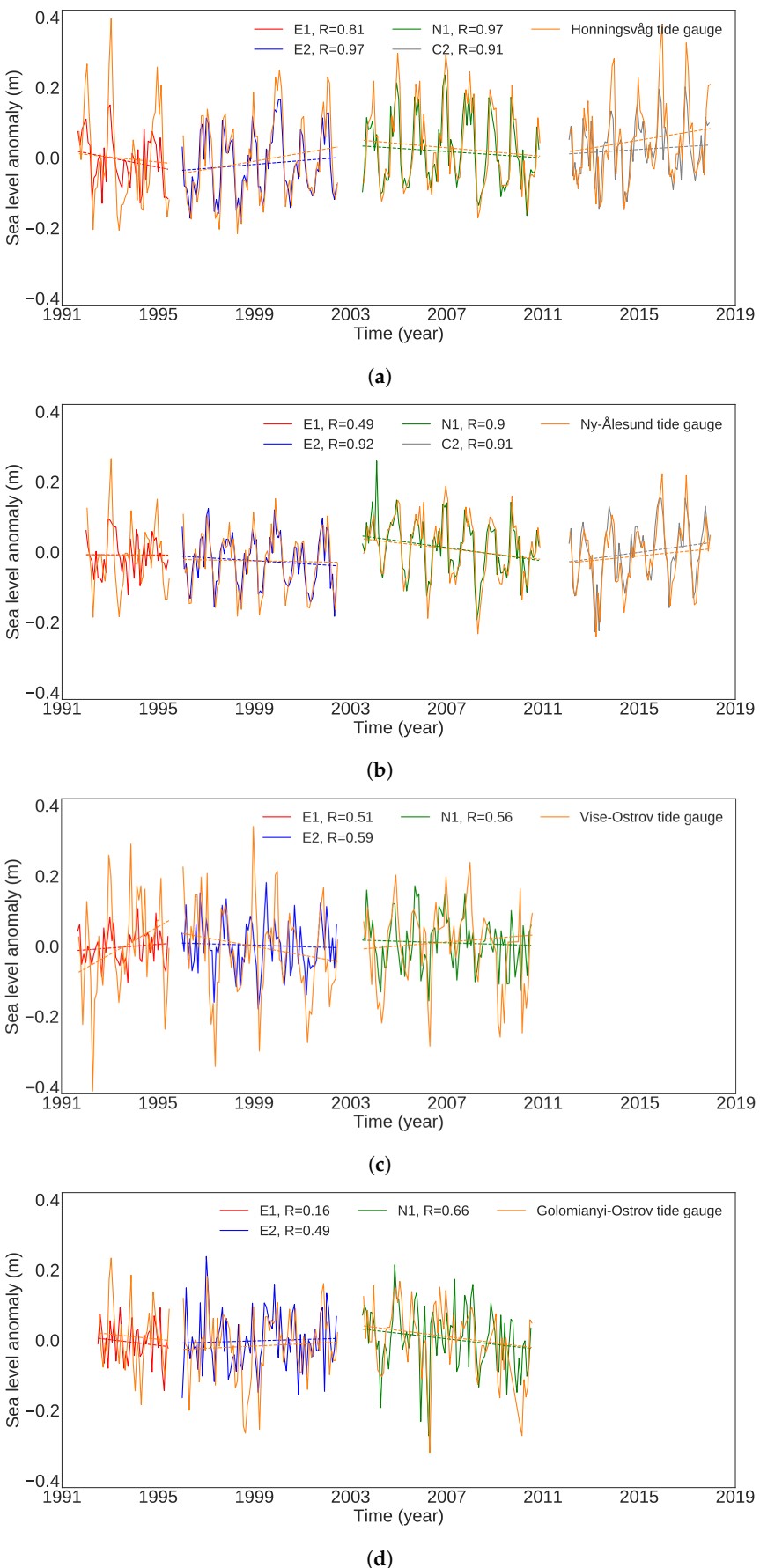

**Figure 8.** *Cont.*

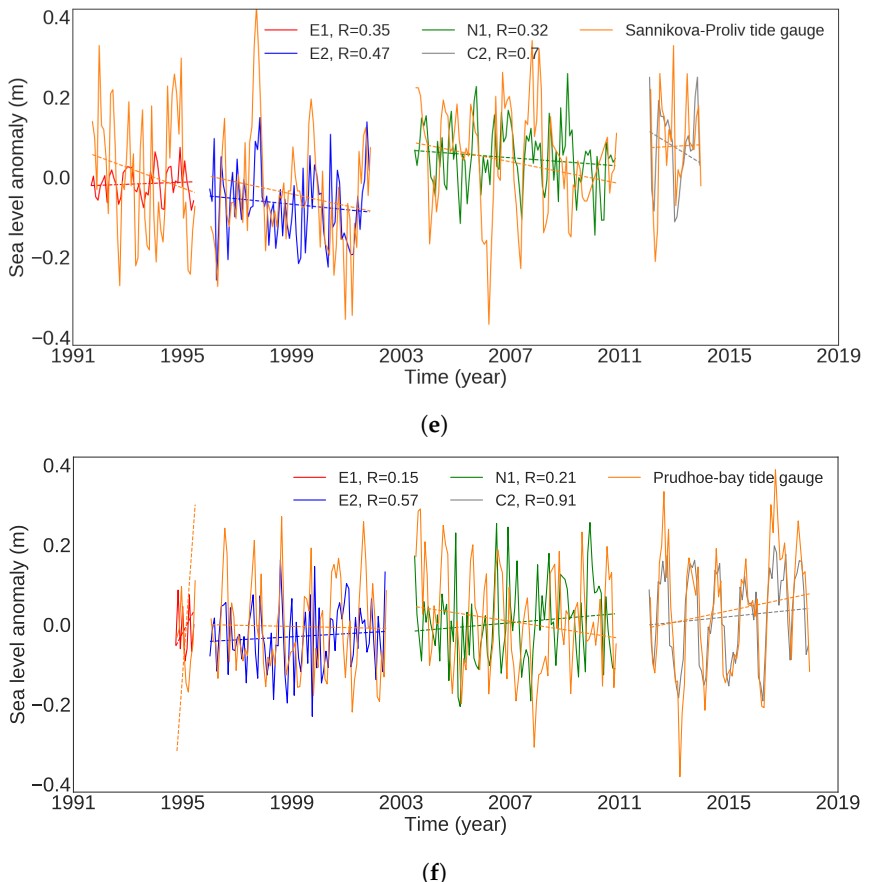

**Figure 8.** Tide gauge comparison of the six tide gauges for each satellite period, where red (ERS-1), blue (ERS-2), green (Envisat) and grey (CryoSat-2) are the altimetric data and orange is the tide gauge data. The tide gauges are evaluated relative to each individual satellite and accordingly a gap appear in between the satellites where data overlaps. Also trend lines are shown on the figures. (**a**) Honningsvåg. (**b**) Ny Ålesund. (**c**) Vise Ostrov. (**d**) Golomianyi Ostrov. (**e**) Sannikova Proliv. (**f**) Prudhoe bay.

**Table 5.** Tide gauge comparisons. The second column shows the number of months analyzed. The second part of the table summarizes the tide gauge comparison given as the RMSE (in meters) with Persons' correlation coefficient in parenthesis for ERS-1 (E1), ERS-2 (E2), Envisat (N1), CryoSat-2 (C2), before GIA correction, in the period starting from 1996 (without ERS-1) and for the total time period.

| Tide Gauges | No. of Month | RMSE (R) | | | | | | |
|---|---|---|---|---|---|---|---|---|
| | | E1 | E2 | N1 | C2 | Pre GIA | 1996– | Total |
| Ny Ålesund | 312 | 0.080 (0.49) | 0.031 (0.92) | 0.034 (0.90) | 0.038 (0.91) | 0.072 (0.70) | 0.042 (0.88) | 0.050 (0.81) |
| Honningsvåg | 316 | 0.089 (0.81) | 0.037 (0.97) | 0.032 (0.97) | 0.060 (0.91) | 0.057 (0.91) | 0.047 (0.94) | 0.055 (0.92) |
| Prudhoe bay | 273 | 0.11 (0.15) | 0.099 (0.57) | 0.15 (0.20) | 0.072 (0.91) | 0.12 (0.53) | 0.11 (0.57) | 0.12 (0.53) |
| Vise Ostrov | 221 | 0.12 (0.51) | 0.10 (0.59) | 0.097 (0.55) | - | 0.11 (0.52) | 0.098 (0.57) | 0.10 (0.53) |
| Golomianyi Ostrov | 202 | 0.099 (0.16 ) | 0.086 (0.49) | 0.077 (0.66) | - | 0.085 (0.53) | 0.081 (0.59) | 0.081 (0.53) |
| Sannikova Proliv | 244 | 0.16 (0.35) | 0.13 (0.47) | 0.13 (0.32) | 0.092 (0.70) | 0.14 (0.36) | 0.14 (0.37) | 0.14 (0.36) |

Table 5 summarizes the results of the comparison. The altimetric SLAs are compared for each satellite and for the entire time series available by the Root Mean Square Error (RMSE) and the Persons correlation coefficient. ERS-1 shows a very good correlation for the Honningsvåg tide gauge, where

there is ice free year round and a good data coverage; a good correlation for Ny Ålesund, fair correlation for Vise Ostrov, weak correlation for Sannikova Proliv and no correlation for Golomianyi Ostrov. In the later tide gauges, we know that the ERS-1 data are very sparse and the tide gauges are placed in a region where sea-ice is changing from season to season. There is a better correlation in the Summer data than in the Winter data, where the ocean is ice free. The correlation for ERS-2, Envisat and CryoSat-2 is excellent for Ny Ålesund and Honningsvåg. For Prudehoe Bay in the Canadian sector the results are also good, except for Envisat where we get a correlation of only 0.21. This is due to loss of data in this area in the Envisat period. Data from the Russian sector are generally having a moderate correlation. There are some very large variation in the tide gauge data (>±0.4 m), which are not captured by the altimetry. This will be discussed in Section 5.3. In Table 5, the last two columns represent the comparisons for the time series without ERS-1 (1996–) and the total time series with ERS-1 (Total). The results improve when the ERS-1 data are ignored.

## 5. Discussion

In this section the results are examined and evaluated.

### 5.1. The SLA Record

ERS-1 was the first radar satellite measuring in the Arctic, and the quality of useful data are sparse, especially in the Beaufort Gyre area. Consequently, several grid cells in this period where empty or close to and therefore not included in the analysis.

It is difficult to assign the quality of the classification in conventional altimetry. In conventional altimetry it is not possible to identify the leads with the same accuracy as in SAR or SARIn. A wrong classification could give a bias with respect to SAR/SARIn. In the transition to SAR/SARIn this could give negative trends and thus an underestimation of the actual Arctic SLA trend. This concern is partly supported by the fact that we generally observe smaller sea level trends in the combined Envisat and CryoSat-2 period compared with the individual Envisat and CryoSat-2 periods and particularly in regions with seasonal sea-ice cover.

In Figure 3, the mean SLA for each satellite were shown. The ice-edge is visible in the subfigures to the East of Svalbard. It is a delicate compromise to keep measurements from conventional satellites in the ice-covered regions or not, as it is impossible to discriminate between reflections from the top of the ice or from leads, causing the average for these satellites to be too high. The chosen PP threshold values may be too loose, but it is a trade-off of either removing some of the signal or getting too many false-positives. We have chosen to go with the second option, and hopefully removing the faulty values in a strict outlier detection. Despite our careful editing, we still find this to be a problem that needs further attention. Getting a more strict PP threshold would lead to areas with very low data coverage, not being able to get a region wide SLA record relaying on data and not only extrapolation. An other option could be to use more advanced classification schemes and machine learning to get a better control of the leads in the sea-ice cover similar to [63–65].

There is an agreement between the four satellites in Figure 3 on positive averaged SLA values in the region (80–82, 0E–100E) north of Svalbard. This might indicate that the DTU18 MSS used to reference the average SLA is too low. CryoSat-2 again shows slightly different signal to the other satellites. This is suspected to be a consequence of the merger of lead data retracked with LARS and open ocean data retracked with RADS. Any discrepancies/offset between these two data sets might cause such a mean signal.

The correlation between ERS-1 and ERS-2 (Figure 2) was moderate. This maybe due to the fact that the ERS-1 satellite's data coverage is poor in both time and space or it could be that the overlapping time period is shorter for this comparison. On the other hand it is very reassuring how well the remaining satellites are matching in the overlapping periods (Figure 2). Especially, it is interesting how well the ERS-2 and Envisat data are correlating when they are processed with the same retracker. Errors in the inter-satellite bias estimation would propagate into the final trend estimation. If the

inter-satellite bias would be wrong, it would also have shown up in the total time series of the tide gauge comparison (See Table 5 and Figure A2 in Appendix D).

Gridded satellite data are very sensitive to the coverage of satellite tracks. In the Arctic, the data distribution is much more dense in the high latitudes when using a constant area grid. Furthermore, the collocation method we are using is very sensitive to missing data and tends to extrapolate data towards zero. Therefore, using this gridding method where there is no or very little data coverage, should be done with caution.

## 5.2. Error Analysis Evaluation

In order to avoid a troublesome error analysis, where the risk of forgetting some error propagation in the process, a bootstrap analysis was carried out. The advantage with bootstrapping is that we do not need to know all the error sources and the size of the uncertainties. The concern with block bootstrapping is to determine the right block size, such that data are independent. The bootstrap method fails if the blocks are too small and we do not get the right variation of data if the blocks are too large. This was tested on trail and error.

Data were not normal distributed (Appendix C). This means that a normal standard deviation evaluation of the uncertainty is not a proper evaluation. In Figure 6, the 95% confidence level was shown. The uncertainty decreases with time with the largest error in the ERS-1 period and smallest for the CryoSat-2 measurements. This is not surprising with the large improvements in the satellites payload giving more data with higher quality.

There is a higher uncertainty in the interior of the Arctic Ocean, than in areas without permanent or seasonal sea-ice cover (Figure 4). Looking at the standard deviation (not shown here—for comparison only) of the bootstrapping realizations the level is between about 2 cm in the interior and about 5 cm outside. This may indicate, that some data are tracked as sea-ice, but the results look similar to Poisson et al. [65] (only using Envisat data) which get a transition between the open ocean and the interior of the Arctic Ocean of about 2 to 4 cm, and much better than the former DTU data set by Cheng et al. [24].

In Figure 7 (Section 4.4), it is striking how low the uncertainty in Sector I is compared to the other sectors. This is the only area with very good satellite coverage, large areas with no sea-ice year round, and with only a little seasonal sea-ice in the northern parts of the sector. The figures clearly indicate that conventional altimetry in the interior of the Arctic Ocean (Sector II, III, IV) before 2010 is more noisy than SAR altimetry from Cryosat-2.

## 5.3. Regional and Seasonal Variability

In the Arctic Ocean a sea level rise of 1.54 mm/year with a 95% confidence interval of 1.16–1.81 mm/year from September 1991 to September 2018 is found. Ignoring the ERS-1 data, a linear trend of 2.22 mm/year with a 95% confidence interval of 1.67–2.54 mm/year from January 1996 to September 2018 is found. These results correspond well with other studies: Cheng et al. [24] used reprocessed RADS data to make the Arctic DTU SLA record V2 and found a SLA trend of $2.1 \pm 1.3$ mm/year in the period 1993–2011. Andersen and Piccioni [26] made an update of the Cheng et al. [24] data and found a trend of $2.2 \pm 1.1$ mm/year in the period 1993–2015. Both studies used data from the sea-ice cover that was processed with an ocean retracker and sampled to 1 Hz. Prandi et al. [23] made an update to the SSALTO/DUACS product from 1993–2009 and found a higher SLA trend of $3.6 \pm 1.3$ mm/year, but they have a very low data coverage in the area corresponding to Sector III.

In Figure 9, the SLA trends are derived by cutting the total time series with five years at a time. Up to 2007 we see an almost constant sea level trend around 1–2 mm/year, but an increased trend can be seen after 2004. There is also evidence, that the loss of multi-year ice is stable up to 2007 where after a steepened loss of multi-year ice is seen [66]. Also the Greenland ice sheet did experience an accelerating ice mass loss already from 2004 [67].

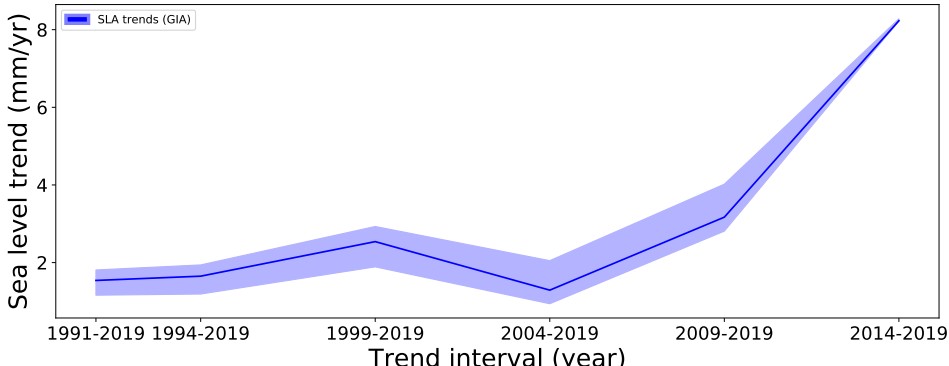

**Figure 9.** Sea level trends in the CCI DTU/TUM SLA period in steps of eliminating five years at a time. The light blue color shows the 95% confidence interval.

In general, the tide gauges show slightly higher sea level variability than the altimetry data (Figures 8 and A2). However, in a few of the tide gauges we suspect that the combination of seasonal sea-ice cover and the location of the tide gauge in sheltered environment away from the harsh Arctic conditions (i.e., up a river) causes the gauge to measure a signal which is smaller than in the open ocean. The local GIA signal can be large, and in Table 4, for the Ny Ålesund tide gauge, we saw how large the difference between the GPS uplift versus the GIA model could be. We also know [59,60] how uncertain the GIA model can be, so applying the GIA correction can be associated with large errors. Sector II was examined in Section 4.4 (Figure 7), Section 4.5 (Figure 8c–e), Appendix D (Figure A2c–e) and Table 5 and found to be the most difficult sector for the altimetry with the worst correlation to the tide gauges.

Inspecting the seasonal variability for the Arctic Ocean in the different sectors (Figure 7), the mean SLA for each month (1996–2019) is plotted (Figure 10) for the entire Arctic Ocean (All) and for each of the geographical sectors (Sector I to IV). We see a maximum SLA for the entire region in October and a minimum in April. This is similar to [25].

There is a minor difference in the seasonal signal in particularly in Sector IV (Figure 7d) which can be explained from the fact that conventional altimetry did not have that many observations in the Canadian Archipelagos, so the seasonal variation was dominated by the seasonal variation in the Baffin Bay and the Beaufort Gyre. This is not the case with the Cryosat-2 data.

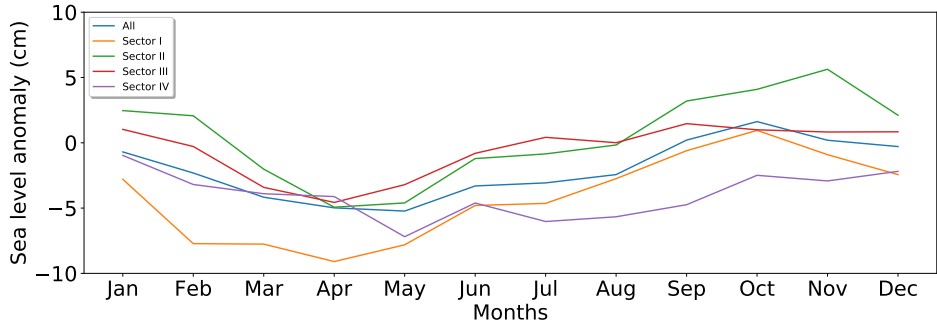

**Figure 10.** The mean value for all months in the interval 1996–2019 for the entire Arctic Ocean (blue) and the different regions outlined in Figure 1: Sector I (orange), II (green), III (red), IV (purple).

The trend pattern in Sector I (Figure 7a) is similar to other studies [23,24,26] with exceptions of the area near the Greenlandic coast, where [26] has a very high SLA trend. For Sector I, we see a maximum SLA in October and a minimum in April, but as described in Volkov et al. [68], the seasonal minimum and maximum can change within small regions. Volkov et al. [68] studied the causes for the sea level variability in this region. The region was divided into smaller areas, and they found a difference in the maximum amplitude (September to December) depending on the area in question (and minimum

in (March to May)). The Barents Sea had a phase lag of 1–3 months compared to the Norwegian and Greenland Sea. This was due to mass changes caused by wind forcing, a varying bottom topography and dissipation. The annual cycles did not change over different time spans.

Sector II (Figure 7b), the Russian Arctic, has a insignificant trend of 0.04 mm/year with a confidence interval of −0.86–1.00 mm/year. This is a product of negative trends particular in the Laptev Sea and positive trends in the Kara Sea and in the East Siberian Sea. In the period before the satellite era 1954–1989 a trend of 1.85 mm/year was found in the Russian sector from in-situ measurements [69], where also negative trends in the inner Kara Sea and in the Laptev Sea were found. 35% of the sea level rise was determined to originate from the ocean expansion, 30% from the atmospheric pressure, 10% from wind forcing and about 25% from increasing ocean mass from melting land ice. A more recent study by Henry et al. [17] covering 1950–2009, also using in-situ observations agrees on these observations. Furthermore, they studied the GIA effect and found that the GIA signal is large in this region and models do not agree well. We do not see the large GIA signal in Figure 7b, but we are looking at a much larger area, where the local effects can vary a lot (see ex. Table 4). The annual SLA variation (Figure 10, green curve) is highest in November and lowest in April.

Sector III (Figure 7c), the Beaufort Gyre region, is the sector with the largest trend (5.77 mm/year with confidence interval 4.12–6.42 mm/year) and with a local maximum trend of 8.45 mm/year. The large trend is due to increasing fresh water accumulation caused by anti-cyclonic winds and Ekman transport [70,71]. There is evidence that the Beaufort Gyre SLA trend has decreased from 1995 to 2003 (−5.9 ± 1.3 mm/year) and steepened from 2003 to 2011 (18.8 ± 0.9 mm/year) [71]. The steepening could also be visible in Figure 7b. Our maximum is smaller, but we are also looking at a different time interval and a larger geographical region. This region has a maximum annual SLA peak (Figure 10, red curve) already in September, a second peak in July and a minimum in April. In a study using moorings [70] from 2003–2007, the authors also find a seasonal cycle with two maximums in June–July and in November–January. They explain the two maximums as originating from the largest yearly sea-ice melt and from the largest wind curl when the salt from sea-ice formation has not jet reached its highest level. Armitage et al. [25] have also studied this area, and evaluated the steric height from 2003–2014. They found a maximum in November, a second peak in June and a minimum in May. This is shifted compared to this study, but it is seen before [68] that the steric and altimetric height not necessarily have the same annual cycle. The GIA effect in this area is large (Figure 7b).

Sector IV (Figure 7d), is an area with very different conditions, having both the archipelagos and the flow through the Baffin Bay. In the archipelagos there are few observations from conventional altimetry. The main signal comes from the Baffin Bay and the CryoSat-2 satellite. There is a seasonal SLA peak in January and a low in May (Figure 10, purple). It is also a region with high GIA values, probable due to melt from the large outlet glaciers in the area. The trend pattern is similar to Carret et al. [72].

## 6. Conclusions

The Arctic Ocean is warming faster than ever, nevertheless the Polar Oceans are not included in global sea level studies due to the uniqueness of the regions and with the associated large errors. In this study, we have presented the CCI DTU/TUM Arctic SLA record including data from four ESA radar altimeter satellites, which is (to current date) the longest time series available. We have carefully combined data from different processings including L2 measurements and state-of-the-art retracking. This can be troublesome, and a lot of cautions have to be taken in combining different time series. Data are validated against six tide gauges spread along the coast in the Arctic Ocean. We found a very good correlation of data in the Fram Strait and a less good correlation in the Russian Arctic due to a bad data coverage. The GIA estimation is uncertain, but when applying the correction, the sea level rise gets larger. A sea level rise of 1.54 mm/year with a 95% confidence interval of 1.16–1.81 mm/year is found in the total time period from September 1991 to September 2018. Ignoring the ERS-1 data and looking

at the period from 1996 to 2018, we get a linear trend of 2.22 mm/year with a 95% confidence interval of 1.67–2.54 mm/year. We handled the troublesome error analysis by a bootstrapping method allowing us to get an uncertainty estimate without keeping track of all the uncertainties in the processing chain.

The trends are associated with relatively large uncertainties. Our trends are likely underestimated in the ice-covered regions of the Arctic, which is a combination of several factors currently under investigation. We had to be tolerant in the editing of the conventional altimetry data in order to get data at all during the ERS-1/ERS-2/Envisat period. The risk is here, that we allow for reflections from the top of the ice, biasing the first part of the time series too high. Vice versa the SAR altimetry from Cryosat-2 over the sea-ice cover is also associated with uncertainties, because we only have SAR data and not SARIn data. This way we are unable to detect off-nadir reflections, which will cause the sea level estimate to be too low in the last part of the time series, and hence the estimated trend will be too low. This could also explain why the altimeteric trend for several stations north of Russia were lower than that observed at the tide gauge.

In several of our results we do seem to see small effects (i.e., Figure 3) related to the behavior of the retracker in the presence of partly to full sea-ice coverage. Biases between data sets processed from different missions and retrackers shall be resolved by cross-calibration, as shown in this study. For Cryosat-2 the merger between RADS used in the open ocean and LARS (in-house gaussian retracker) used in the sea-ice seems to result in smaller seasonal SSH effects. The latter effect is particularly hard to tackle as the data are disjoint to each other. This is currently work in-progress to improve the consistency among the different retrackers.

Monthly gridded SLA maps from September 1991 to September 2018 is available at https://ftp.sp ace.dtu.dk/pub/ARCTIC_SEALEVEL/DTU_TUM_V3_2019/ and at ESA SL_CCI (cci.esa.int/data).

**Author Contributions:** Conceptualization and methodology: S.K.R., O.B.A., software, formal analysis, investigation, writing—original draft preparation: S.K.R., validation: S.K.R., O.B.A., C.A.L., data curation: S.K.R, M.P. (ALES+ retracking), ALES+ data storage and organization: C.S., supervision, funding acquisition: O.B.A. M.P., writing—review and editing: S.K.R., O.B.A., M.P.

**Funding:** This research was funded by ESA Climate Change Initiative Sea level Budget Closure, contract No. 4000119910/17/I-NB.

**Acknowledgments:** The authors thank Benjamin D. Gutknecht for providing the VLM grids. The first author also acknowledge Karina Nielsen for fruitful discussions and recommendations in the bootstrap analysis. Furtmore, the authors are gratefull to the reviewers for constructive and positive review.

**Conflicts of Interest:** The authors declare no conflict of interest.

## Abbreviations

The following abbreviations are used in this manuscript:

| | |
|---|---|
| ALES | Adaptive Leading Edge Subwaveform |
| AO | Arctic Oscillation |
| DAC | Dynamic Atmosphere Correction |
| DTU | Technical University of Denmark |
| ECMWF | European Centre for Medium-Range Weather Forecasts |
| ESA | European Space Agency |
| ERS | European Remote Senesing satellite |
| GIA | Glacial Isostatic Ajustment |
| IBE | Inverse Barometer Effect |
| IPCC | Intergovernmental Panel on Climate Change |
| LRM | Low Resolution Mode |
| MAD | MediAn Deviation |
| MSS | Mean Sea Surface |
| PP | Pulse Peakiness |
| PSMSL | Permanent Service for Mean Sea Level |
| RADS | Radar Altimetry Database System |

| REAPER | Reprocessing of Altimeter Product for ERS |
|---|---|
| SSH | Sea Surface Height |
| SAR | Synthetic Aperture Radar |
| SARIn | SAR Interferometry |
| SGDR | Sensor Geophysical Data |
| SLA | Sea Level Anomaly |
| SLBC_CCI | Sea Level Budget Closure Climate Change Initiative |
| SL_CCI | Sea Level Climate Change Initiative |
| TUM | Technical University of Munich |

## Appendix A. Satellite Specific Processing Details

### Appendix A.1. ERS-1

ERS-1 was launched in July 1991 carrying among other instruments a pulse-limited single frequency $K_u$ band (13.8 GHz) Radar Altimeter (RA). RA measured with a footprint of 16–20 km spatial resolution and with an accuracy of 10 cm over the ocean. It was the first Earth observing ESA satellite with a Sun-synchronous polar orbit (inclination: 98.52°) allowing measurements up to about 81.5°. ERS-1 had a repeat cycle of: 3-days, 35-days and 176-days. The mission failed in March 2000, but already in 1996 the ERS-2 satellite (launched April 1995) took over the operational services [73]. For ERS-1 we use the Reprocessing of Altimeter Products for ERS (REAPER) [34] L2 data set.

There are known orbital errors for the ERS satellites as a consequence of lacking accuracy of gravity data and International Terrestrial Reference Frame (ITRF) realizations. It was found necessary to correct for orbital errors even though one of the REAPER project scopes was to make a better orbit solution [34]. For ERS-1, we use a orbit correction scheme similar to [74], described in more details in Cheng et al. [24]. This study is deviating from [74] by not aligning data to the TOPEX/Jason-1/2 SLA because the coverage of the satellites only reached up to 66°N. To get rid of the most noisy measurements, abnormal outliers (>10 m) are removed and a median deviation (MAD) outlier detection is applied to each track before further analysis.

### Appendix A.2. ERS-2

ERS-2 was launched in the same orbital plan as ERS-1, but with a one day lag allowing for tandem measurements. ERS-2 was launched with almost identical instruments as ERS-1 with a few improvements. ERS-2 was sending valuable measurements back to the ground station till June 2003 and failed entirely in 2011 [73]. For ERS-2 we use the ESA Sensor Geophysical Data Records (SGDR) of ERS-2 REAPER [34] covering the period from September 1995 to July 2003.

### Appendix A.3. Envisat

Envisat is ERS-2's successor. It was launched in March 2002, as the largest ever built satellite with 10 different instruments aboard. ESA lost contact to Envisat in May 2012. The radar altimeter on-board (RA-2) was a pulse-limited dual-frequency radar operating in the $K_u$ (13.575 GHz) band and S bands. Only the $K_u$ band is used here. The spatial resolution is 2–10 km [11] with an accuracy better than 4.5 cm. The satellite's turning latitude was 82° [73]. Envisat had a repeat cycle of 35-days [73]. The SGDR Envisat version 2.1 is used. For Envisat the entire duration of the phase 2 (May 2002—October 2010) and phase 3 (November 2010–May 2012) is used.

The ERS-2 and Envisat satellites are processed with the ALES+ retracker [35]. It is an upgraded version of the Adaptive Leading Edge Subwaveform (ALES) Retracker [75] that is a retracker adapted to coastal ocean areas, without lowering the quality of the results in the open ocean. ALES+ is developed to improve retrievals of peaky waveforms such as echos from leads in the sea-ice cover. Particularly, one large advantage of this retracker is the seamless transitions between leads and open ocean waveforms. For unknown reasons data is missing in ERS-2 covering the weeks: 15 April 2000 to 7 May 2000, 2 July 2000 to 9 July 2000, 21 January 2001 to 4 February 2001,

10 March 2002 to 17 March 2002. For Envisat data are missing in the weeks: 16 March 2003 to 23 March 2003 and 22 March 2011 to 29 March 2011.

*Appendix A.4. CryoSat-2*

Cryosat-2 was launched on April 2010 and is still active. CryoSat-2 is a dedicated cryosphere satellite with a coverage up to 88° latitude measuring more of the Arctic than ever before. It has a 369-day repeat cycle. The main instrument of CryoSat-2 is the $K_u$ (13.6 GHz) band SIRAL-2 (SAR/Interferometric Radar Altimeter-2). SIRAL is able to measure in one of three modes; LRM over ocean and flat surfaces such as the interior of the ice sheets; SAR mode mainly over sea-ice covered ocean; SARIn over steep terrain and coastal areas. We will be using data from all three modes. The conventional LRM is a pulse-limited footprint, in SAR mode the Doppler principle results in a narrow along-track footprint which can be seen as a beam-limited footprint. The SARIn mode utilizes the two antennas on CryoSat-2 forming an across-track interferometer. The echoes received by each antenna undergo Doppler beam processing as in SAR mode, but the number of waveforms averaged is lower due to the longer interval between the bursts. Processing with multi-looks results in a waveform with a more sharpened leading edge and stronger peak power [73]. For SAR and SARIn baseline C Ice level 1B data are used.

The CryoSat-2 data contains LRM, SAR and SARIn. For LRM and SAR the 1 hz Radar Altimetry Database System (RADS) [38] are used. The specular lead returns from the 20 hz SAR and SARIn are retracked using the Lars Advanced Retracking System (LARS) [37] using a Gaussian fitting routine similar to [13]. Off-nadir SARIn data are processed as SAR data. For unknown reasons almost one month of SAR/SARIn data is missing from 12 August 2010 to 16 September 2010.

**Appendix B. The Ice Concentration Grid**

In each ice concentration (Section 2.2) grid cell a percentage of the ice concentration is given. A threshold greater than 15% is defined as a cell with sea-ice, and if the cell is below 15% the cell is classified as open ocean. The ice concentration masks are also used to remove CryoSat-2 RADS data from the ice cover, such that the RADS data is only used over the open ocean.

Every satellite point is tracked in the sea-ice concentration grid and evaluated in terms of its location with respect to the sea-ice cover. The satellite coordinate is tracked by a *k*-dimensional (kd)-tree for quick nearest-neighbor lookup for the closest coordinate.

**Appendix C. SLA Distributions and Uncertainty Estimates**

For each monthly data set, 1000 new bootstrap realizations are generated by splitting data in *n* non-overlapping blocks as done in the making of the SLA (Section 3.6). The bootstrapping blocks need to be independent from each other, hence the size of the blocks was tested. This test was done as a trial and error. When the block sizes were too small the bootstrapping failed. Furthermore, it was a wish to get so much variation in the data as possible, hence the block size should be as small as possible. The final block sizes are three times the size of the first averaging i.e., $0.6° \times 0.6°$ in the latitude and longitude direction. The bootstrapping is carried out by randomly drawing *n* blocks with replacements from the SLA data set. In practice, this means that some blocks are appearing more times and some are not represented at all. For each monthly grid cell, for all 1000 bootstrap realizations, a new SLA is calculated as done in the previous section, and a 95% confidence level is estimated for every grid cell.

The SLAs in each grid cell in the Arctic Ocean is not normal distributed. Figure A1 shows the monthly SLA distribution of the median of the 1000 bootstrap realizations for selected grid cells distributed in the Arctic Ocean. It is only valid to show results with a standard deviation if the results are normal distributed. The Arctic SLA distributions are not normal distributed for all grid cells in the Arctic Ocean, therefore the uncertainty is expressed in a 95% confidence level. This was also verified by q-q plots (not shown here). Therefor we use 95% confidence level as the uncertainty estimate.

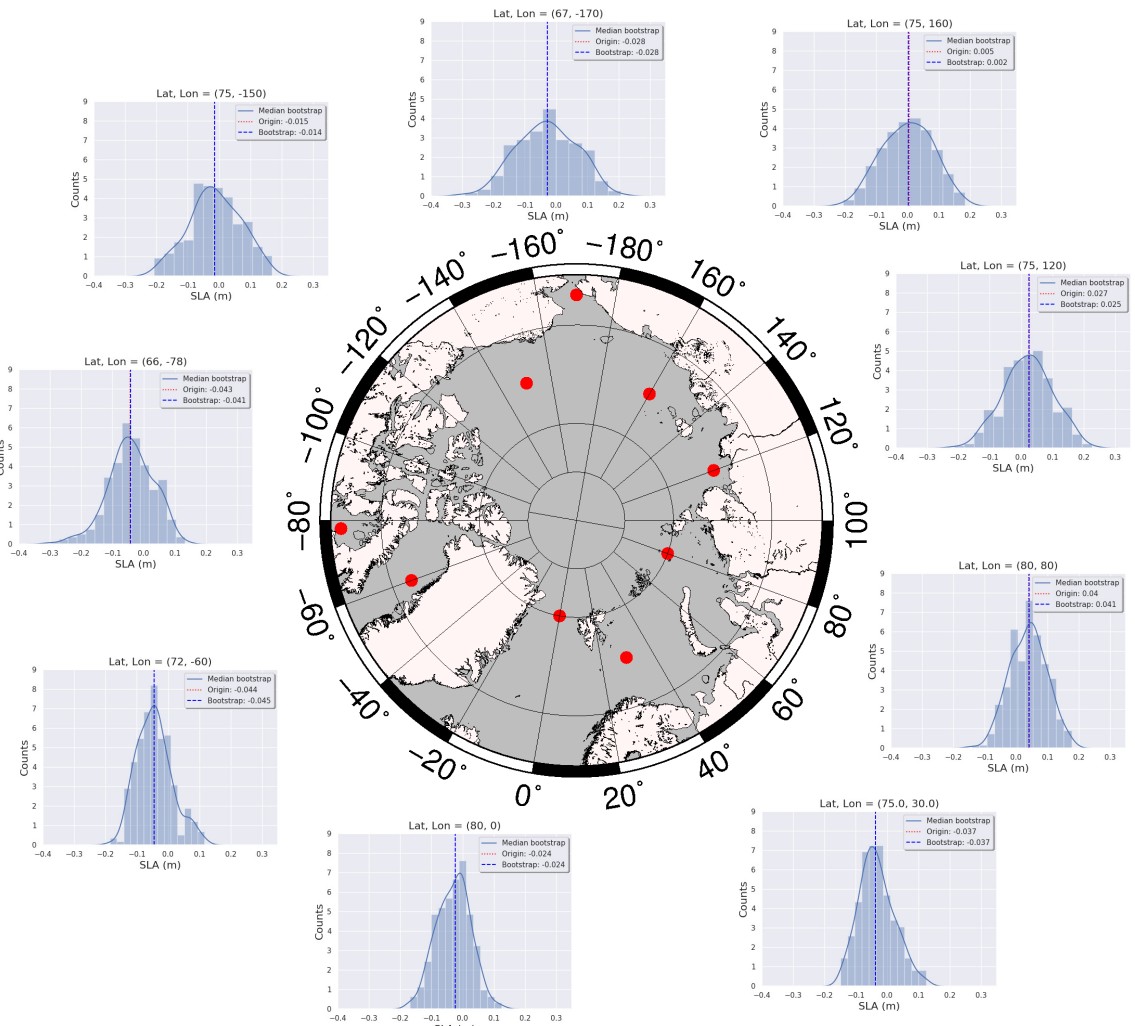

**Figure A1.** Examples of median bootstrap SLA distribution from January 1991 to September 2018 for various grid points. The vertical lines represent the median of the original SLA (red) and the median bootstrap realizations (blue).

## Appendix D. Tide Gauge Comparison

The altimetry data were average with a radius of 350 km around te tide gauge with an inverse distance weighted function. Various radii around the tide gauges and a simple median average were tested, but this method showed the best results for the overall result. For all tide gauges, except for Ny Ålesund, the inverse distance weighting of data (not shown) improves the results. This may indicate that there is control of the outliers. The reason why Ny Ålesund does not improve by the inverse distance weighting, could be due to the fact that the tide gauge is situated on an island, where there are a lot of coastal area with many fjords. To make this point stronger it is also ERS-1 and ERS-2 that suffer the most when doing the weighted mean. The worse performance of ERS-1 and ERS-2 near the coast are due to the lower pulse repetition frequency of the satellites. A simple median average of the data would give a better solution in areas with little data ex. for ERS-1 and in the Russian Arctic. The amplitude of the altimeter data gets larger for the weighted solution and also the trends get closer. The tide gauge in the Laptev Sea (Sannikova Proliv) performs best when eliminating data with an interpolation error >0.10 (not shown). This is normal an indicator of missing or very low data coverage.

In Figure A2 the total altimetry time series (blue) within 350 km of the tide gauge is compared with the tide gauge data (orange). Also the trend lines are shown and the different trends are written in the figure caption. The uncertainties are given by three times the standard deviation.

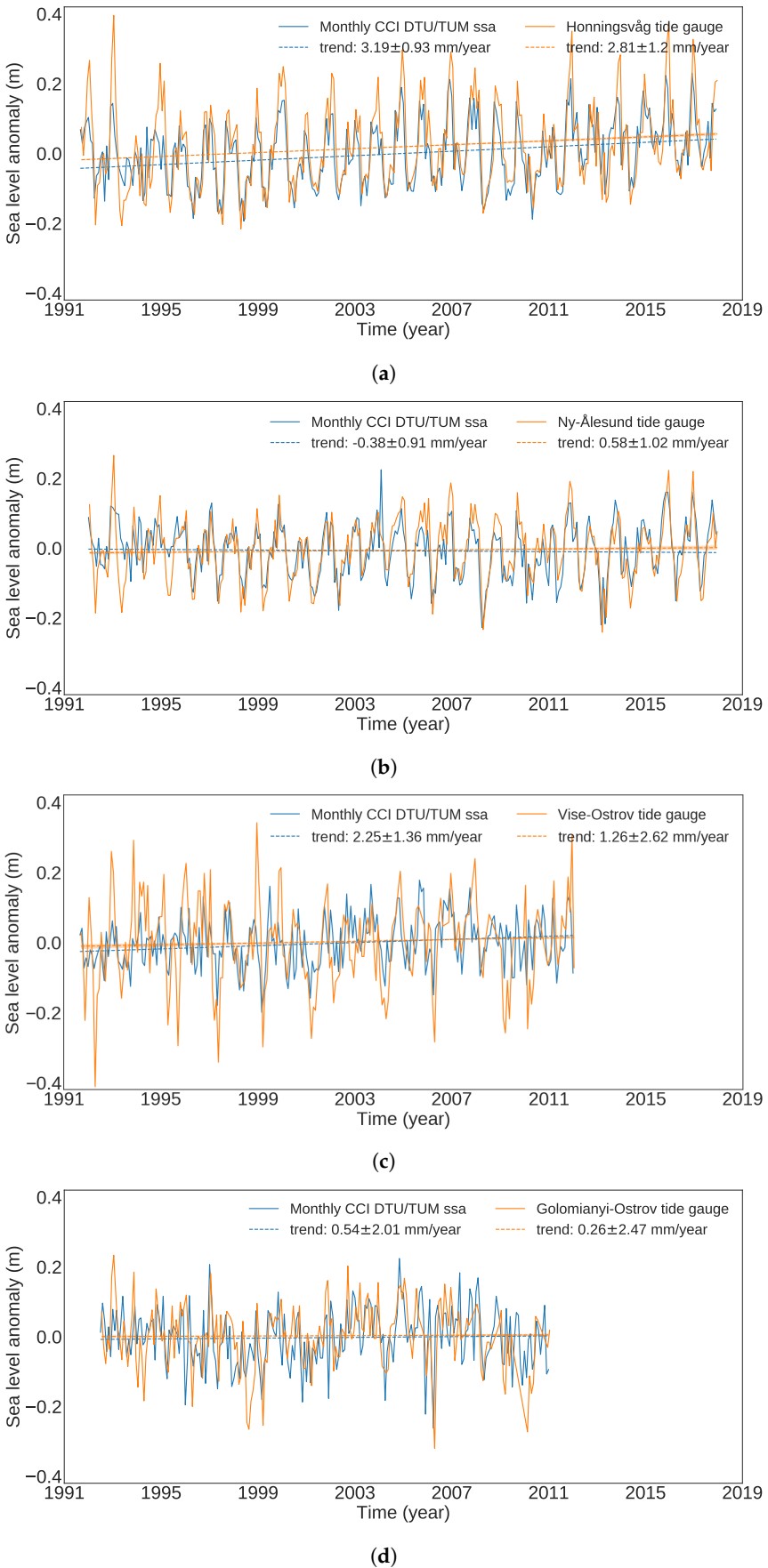

(**a**)

(**b**)

(**c**)

(**d**)

**Figure A2.** *Cont.*

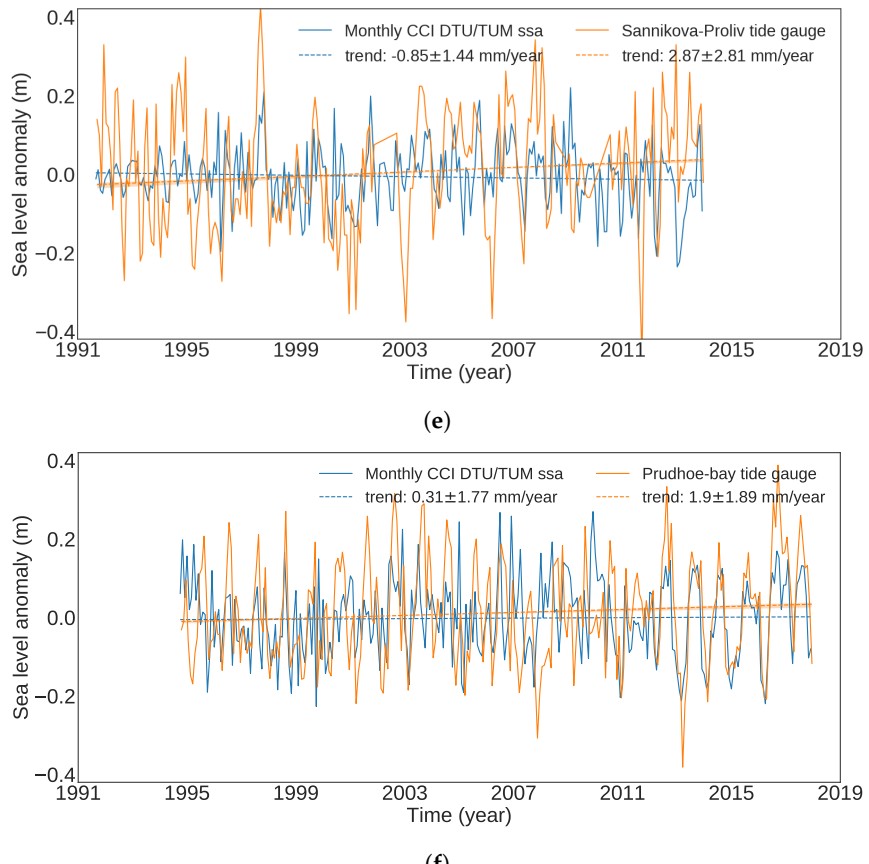

**Figure A2.** Tide gauge comparison of the continued altimetry time series from the six tide gauges in the period from 1991 to 2019, where blue is the altimetry data and orange is the tide gauge data. The vertical lines are the trend lines colored like the time series. The various trends are written in the captions. (**a**) Honningsvåg. (**b**) Ny Ålesund. (**c**) Vise Ostrov.(**d**) Golomianyi Ostrov.(**e**) Sannikova Proliv.(**f**) Prudhoe bay.

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
