# Peer review of "Arctic Ocean Sea Level Record from the Complete Radar Altimetry Era: 1991–2018"

_remotesensing, doi:10.3390/rs11141672_

Round 1
Reviewer 1 Report
All my questions have been explained. I recommend the publication of this paper at Remote Sensing.
Author Response
Thanks
Reviewer 2 Report
Arctic Ocean Sea Level Record from the Complete Radar Altimetry Era: 1991-2018
Stine Kildegaard Rose, Ole Baltazar Andersen, Marcello Passaro, Carsten Ankjær Ludwigsen and Christian Schwatke
The paper presents an important contribution to altimetry data for the Arctic region. Data of four missions are used for constructing a consistent data set covering all the altimetry era in the Arctic. The authors use a wide range of tools to check the data for errors and uncertainties and to derive estimates of interannual trends of sea surface height and sea surface anomalies in the region. The data are related with in situ measurements by tide gauges. The paper is full of unnecessary details that make it rather hard to follow.
The referee recommends a major revision of the paper before its publication. The focus could be made on the text clarity and avoiding too many details. The latter can be reached by moving some parts to Appendices.
Minor remarks are given below.
218 dimentional -> dimensional;
220 "backscatter" is likely NRCS - normalized radar cross-section;
233 The alignment with the TP data looks questionable because TP does not cover polar regions. This point requires comments;
Figure 2 caption - Please, capitalize Arctic;
236 Low value of the correlation of ERS-1/ERS-2 is likely explained by short series: two times less than for other pairs;
238-240 The phrase seems incorrect;
244-245 Where this bias -12.9 cm comes from? Please, give a reference;
249 Why the outlier cutoff 0.3m is accepted? Please, provide the percentage of the outliers or other arguments for this choice;
3.7. Error Analysis. The referee does not see the error analysis but quite a general description of the bootstrapping approach. What are the sources of errors? What are their values?
Fig.3 shows very different distributions for satellite missions. There are no comments in the text;
Fig.10 and 11 have the same captions but likely illustrate different features of gauge data. Symbols and color lines are not explained.
381-386 What these figures talking about? How the authors improve and substantiate the choice of threshold values?
395 extend -> extent;
419 are-> is;
Sect.5.2. Error analysis issues. What are these issues? Please, formulate with proper emphasis.
467 sow-> show
Author Response
@page { margin: 0.79in } p { margin-bottom: 0.1in; line-height: 120% }Dear Reviewer,
We thank you for the valuable comments. We believe they have helped us to improve the manuscript.
We have answered the comments in red font, and the reviewer's comments are in black.
The paper presents an important contribution to altimetry data for the Arctic region. Data of four missions are used for constructing a consistent data set covering all the altimetry era in the Arctic. The authors use a wide range of tools to check the data for errors and uncertainties and to derive estimates of inter-annual trends of sea surface height and sea surface anomalies in the region. The data are related with in situ measurements by tide gauges. The paper is full of unnecessary details that make it rather hard to follow.
The referee recommends a major revision of the paper before its publication. The focus could be made on the text clarity and avoiding too many details. The latter can be reached by moving some parts to Appendices.
We have made an Appendix as the reviewer suggested and moved several parts to this, especially the more detailed method sections are moved. We have reorganized several sections, so hopefully, this will make the manuscript easier to follow.
Minor remarks are given below.
218 dimentional -> dimensional;
Done
220 "backscatter" is likely NRCS - normalized radar cross-section;
Changed to backscatter coefficient.
233 The alignment with the TP data looks questionable because TP does not cover polar regions. This point requires comments;
It should have been the TP ellipsoid. We have removed this bullet point, because it is not relevant with absolute values in this study. Instead the offset between the satellites are written down, which could be of interest for some readers.
Figure 2 caption - Please, capitalize Arctic;
Done
236 Low value of the correlation of ERS-1/ERS-2 is likely explained by short series: two times less than for other pairs;
True! This is added to the discussion. L370-371
238-240 The phrase seems incorrect;
We have changed the sentences.
244-245 Where this bias -12.9 cm comes from? Please, give a reference;
This is a value found in this study. We have change the formulation of this paragraph.
249 Why the outliers cutoff 0.3m is accepted? Please, provide the percentage of the outliers or other arguments for this choice;
We have change 0.3m to 0.4m. This was a large mistake of the first edition of the manuscript. Thanks for forcing us to look at the scripts again! We have added the percentages and a reference. For ERS-1 we have removed almost 20% of data. This is a consequence of bad data sampling giving bad waveforms. Moreover the corrections from this time is erroneousness. For the remanding satellites the percentage is below 3%.
3.7. Error Analysis. The referee does not see the error analysis but quite a general description of the bootstrapping approach. What are the sources of errors? What are their values?
We assume that what you are seeking for is 1.) a list of the possible error sources and 2.) that the text should be more specific. The authors have tried to rewrite the section to make it more clear. This section was meant to be a description of the method - we have changed the section name. We have written the error sources in the text, and given a reference for the size of the error sources. The errors are assumptions and many not well defined. This is one of the main reasons for making the bootstrap analysis. With this analysis we do not need to know the size of the error sources, because the method contains all the variation coming from the error sources, as long as our blocks are independent. Of course this is a critical point! Are they independent? We want the blocks to be as small as possible to get as much variation as possible, but large enough to be independent. This has been tested on trail and error. Block bootstrapping is a well know method in statistics and highly accepted. We believe this is a much more correct method to describe the uncertainties than a regular method used in satellite altimetry by taking n*std of the result.
Fig.3 shows very different distributions for satellite missions. There are no comments in the text;
This has been commented now. See L: 215-220, 364-368
Fig.10 and 11 have the same captions but likely illustrate different features of gauge data. Symbols and color lines are not explained.
Fig 10 is added to appendix. The captions for both figures are updated.
381-386 What these figures talking about? How the authors improve and substantiate the choice of threshold values?
This is not very relevant for the manuscript., so the paragraph has been deleted to eliminate some of the details.
395 extend -> extent;
Dome
419 are-> is;
Done
Sect.5.2. Error analysis issues. What are these issues? Please, formulate with proper emphasis.
Texts are added to the section and title changed
467 sow-> show
Done
Round 2
Reviewer 2 Report
Dear Authors,
Thank you for your hard work in improving the paper. The referee hopes it will find its grateful reader. The referee is satisfied by key corrections made but does not guarantee the absence of possible misprints and minor omissions. Please, check it once more.
Figs.8 and A2 should be enlarged in the final version.
Sincerely yours,
Referee 4
This manuscript is a resubmission of an earlier submission. The following is a list of the peer review reports and author responses from that submission.
Round 1
Reviewer 1 Report
The authors have conducted a systematic study and analysis of sea level changes in the Arctic Ocean using satellite altimetry data from multiple missions from 1991 to 2018. The manuscript is in general well organised though there sre some issues to be addressed as mentioned in my review comments. It is my view that the paper should not accepted until authors address all issues.

Reviewer 2 Report
The authors released Arctic Sea Level products based on ERS-1/2, Envisat and Cryosat-2 data over the time period between 1991 and 2018. The products were developed using the updated geophysical corrections and a combination of altimeter data: Reprocessing of Altimeter Product for ERS (REAPER) (ERS-1), ALES+ retracker (ERS-2, Envisat), combination of Radar Altimetry Database System (RADS) and DTUs in-house retracker LARS (CryoSat-2). Then the regional sea level trends and variability were investigated and concluded a sea level rise of 1.54 mm/yr during the study period. Moreover, the authors found a clear steepening of the SLA trend around 2004 and an accelerated sea level rise in the Arctic Ocean.
In general, the paper is well organized and written. Several issues need to be clarified:
(1) It seems that the sea level rise acceleration is more significant in sector 1. The acceleration rate could be provided in the various sectors.
(2) Check the time span in the figure captions, it should be up to 2018 (April) in figures 4 and 9.
(3) The altimetry data show higher sea level variability than tide gauge data. It should be explained (Figure 9).
(4) Figure 11, why the cryostat time series is ‘smoother’ than ERS/Envisat data in the Arctic Ocean?
Reviewer 3 Report
General comments:
The paper describes a new sea level dataset for the Arctic Ocean.
I think this new dataset is a nice addition to the current landscape of Arctic Ocean sea level studies, and
a proper description of data and processing methods is welcome.
This does cover the full radar altimeter record, including high rate ERS-1 data, which is a nice effort.
However the paper has a number a inaccurate statements.
It is also at times hard to read: some sections feel really short, while others convey too many details that are not necessary.
The error analysis is not well described and therefore not convincing.
Some conclusions are not supported by the evidence presented in the paper.
I recommend a major review.
Detailed comments (according to line numbers):
27. I dont't see how GHG accumulating in the ocean are linked to its heat content increase,
28. I'm no expert on this topic but I doubt heat content is expressed in W/m2
51. 2 km diameter feels like a very small footprint for a LRM radar altimeter
56-57. off nadir returns were already adressed a few lines before
61-65. maybe mention T/P and the Jason series here
71. Argo floats do not work well in the Arctic Ocean, however other sources of in-situ data are available, such as ITPs for example
106. depending on the mission, different level of details are given, which puzzles the reader : why is the Envisat PRF a relevant information, and why don't we have a similar info for ERS-2 ? for example
121. for CryoSat-2 data, data could come from different ground processors (ICE vs OCEAN), these differ at L1 level and which is used should be made clear
159. the ALES+ retracker does not improve waveforms, it can improve retrieval of geophysical parameters from the waveform
Table 1. references should be provided for each correction,
191. MOG2D refers to the barotropic response to wind and pressure model, end ERA-Interim to the forcing, so there is no DAC MOG2D, its either DAC-ERA Interim or DAC-ECMWF
212. please clarify the meaning of 'tracked' here
Figure 3. on the ERS-1, 2 and Envisat maps, a step is visible roughly were the maximum ice edge is, suggesting that echoes from the top of the ice are considered as sea level measurements, CryoSat-2 shows a very different pattern (looks anti-correlated) do you have an explanation for this ?
280. The section is unclear to me. What are the bootstrap samples drawn from, along-track data ? or 0.2° averages ? how did you check for independence between samples ? Moreover the fact that any systematic error would go unnoticed by this procedure is not discussed.
344-346. This is not supported by the figures in Table 4. where the total period shows higher RMSE and lower R
Figure 11. For all sectors of the Arctic interior, it seems that there is a change in the seasonal cycle when switching to CryoSat-2 data. Do you have an explanation for this behavior ?
460. what do you mean by 'ceased to be the most difficult' here ?
473. you write that the SLA decreased in the Beaufort Gyre, but cite a positive trend number to support that...
502. I understood that the gridded product resulted from an interpolation scheme (described in section 3.3.3), here you are making a reference to median SLA grids, which are not described.